# Aminoglycoside tolerance in *Vibrio cholerae* engages translational reprogramming associated with queuosine tRNA modification

**Louna Fruchard[1,2†], Anamaria Babosan[1†], Andre Carvalho[1], Manon Lang[1], Blaise Li[3], Magalie Duchateau[4], Quentin Giai Gianetto[4,5], Mariette Matondo[4], Frederic Bonhomme[6], Isabelle Hatin[7], Hugo Arbes[7], Céline Fabret[7], Enora Corler[7], Guillaume Sanchez[8], Virginie Marchand[8], Yuri Motorin[8], Olivier Namy[7], Valérie de Crécy-Lagard[9,10], Didier Mazel[1*], Zeynep Baharoglu[1*]**

[1]Institut Pasteur, Université Paris Cité, Unité Plasticité du Génome Bactérien, Paris, France; [2]Sorbonne Université, Collège Doctoral, Paris, France; [3]Institut Pasteur, Université Paris Cité, Bioinformatics and Biostatistics Hub, Paris, France; [4]Institut Pasteur, Université Paris Cité, Proteomics Platform, Mass Spectrometry for Biology Unit, Paris, France; [5]Institut Pasteur, Université Paris Cité, Department of Computation Biology, Bioinformatics and Biostatistics Hub, Paris, France; [6]Institut Pasteur, Université Paris cité, Epigenetic Chemical Biology Unit, Paris, France; [7]Université Paris-Saclay, CEA, CNRS, Institute for Integrative Biology of the Cell (I2BC), Gif sur Yvette, France; [8]Université de Lorraine, CNRS, Inserm, UAR2008/US40 IBSLor, Epitranscriptomics and RNA Sequencing Core Facility and UMR7365 IMoPA, Nancy, France; [9]Department of Microbiology and Cell Science, University of Florida, Gainesville, United States; [10]University of Florida Genetics Institute, Gainesville, United States

*For correspondence:
didier.mazel@pasteur.fr (DM);
zeynep.baharoglu@pasteur.fr
(ZB)

†These authors contributed
equally to this work

Competing interest: The authors
declare that no competing
interests exist.

Reviewing Editor: Srujana
Samhita Yadavalli, Rutgers, The
State University of New Jersey,
United States

## eLife Assessment

This study investigates the role of queuosine (Q) tRNA modification in aminoglycoside tolerance in *Vibrio cholerae* and presents **convincing** evidence to conclude that Q is essential for the efficient translation of TAT codons, although this depends on the context. The absence of Q reduces amino-glycoside tolerance potentially by reprogramming the translation of an oxidative stress response gene, rxsA. Overall, the findings point to an **important** mechanism whereby changes in Q modification levels control the decoding of mRNAs enriched in TAT codons under antibiotic stress.

**Abstract** Tgt is the enzyme modifying the guanine (G) in tRNAs with GUN anticodon to queuosine (Q). *tgt* is required for optimal growth of *Vibrio cholerae* in the presence of sub-lethal aminoglycoside concentrations. We further explored here the role of the Q34 in the efficiency of codon decoding upon tobramycin exposure. We characterized its impact on the overall bacterial proteome, and elucidated the molecular mechanisms underlying the effects of Q34 modification in antibiotic translational stress response. Using molecular reporters, we showed that Q34 impacts the efficiency of decoding at tyrosine TAT and TAC codons. Proteomics analyses revealed that the anti-SoxR factor RsxA is better translated in the absence of *tgt*. RsxA displays a codon bias toward tyrosine TAT and overabundance of RsxA leads to decreased expression of genes belonging to SoxR oxidative stress

regulon. We also identified conditions that regulate *tgt* expression. We propose that regulation of Q34 modification in response to environmental cues leads to translational reprogramming of transcripts bearing a biased tyrosine codon usage. In silico analysis further identified candidate genes which could be subject to such translational regulation, among which DNA repair factors. Such transcripts, fitting the definition of modification tunable transcripts, are central in the bacterial response to antibiotics.

## Introduction

Antimicrobial resistance is an increasingly serious threat to global public health. Our recent finding that many tRNA modification genes are involved in the response to antibiotics from different families (*Babosan et al., 2022*) led to further investigate the links between environmental factors (e.g. traces of antibiotics), tRNA modifications, and bacterial survival to antibiotics.

The regulatory roles of RNA modifications were first proposed for eukaryotes (*Pollo Oliveira and de Crécy Lagard, 2019*) and their importance in human diseases has recently emerged (*Chujo and Tomizawa, 2021*; *Suzuki, 2021*). In bacteria, while some tRNA modifications are essential (*Zhong et al., 2019*), the absence of many RNA modification shows no growth phenotype in unstressed cells (*de Crécy-Lagard and Jaroch, 2021*). At the molecular level, the roles of tRNA modifications in differential codon decoding have been described in various species (*Bruni et al., 1977*; *Parker, 1982*; *Taylor et al., 1998*; *Urbonavicius et al., 2001*). In most cases, no growth phenotype was associated with these variations in decoding in bacteria. Recent studies, however, do highlight the links between tRNA modifications and stress responses in several bacterial species (*Aubee et al., 2016*; *Chionh et al., 2016*; *de Crécy-Lagard and Jaroch, 2021*; *Fleming et al., 2022*; *Hou et al., 2017*; *Thompson and Gottesman, 2014*; *Thongdee et al., 2019*; *Večerek et al., 2007*), and new modifications are still being discovered (*Kimura et al., 2020*). Until recently, few tRNA modification factors have been clearly linked with resistance and persistence to antibiotics, via differential codon decoding in cell membrane and efflux proteins (TrmD [*Masuda et al., 2019*], MiaA [*Taylor et al., 1998*]). A link between stress and adaptation was described to occur via the existence of modification tunable transcripts (MoTTs).

MoTTs were first (and mostly) defined in eukaryotes as transcripts that will be translated more or less efficiently depending on the presence or absence of tRNA modifications (*Endres et al., 2015*), namely upon stress (*Advani and Ivanov, 2019*). In bacteria, links between tRNA modifications and the response to several stresses are highlighted by studies focusing on the following MoTT/codon and tRNA modification couples reviewed in *de Crécy-Lagard and Jaroch, 2021*: differential translation of RpoS/leucine codons via MiaA (*Escherichia coli*) (*Aubee et al., 2016*); Fur/serine codons via MiaB, in response to low iron (*E. coli*) (*Večerek et al., 2007*); MgtA/proline codons via TrmD, in response to low magnesium *Hou et al., 2017*; catalases/phenylalanine and aspartate codons via TrmB, during oxidative stress (*Pseudomonas aeruginosa*) (*Thongdee et al., 2019*). Mycobacterial response to hypoxic stress (*Chionh et al., 2016*) also features MoTTs. In this latter study, specific stress response genes were identified in silico, through their codon usage bias, and then experimentally confirmed for their differential translation.

During studies in *V. cholerae*, we recently discovered that t/rRNA modifications play a central role in response to stress caused by antibiotics with very different modes of action (*Babosan et al., 2022*), not through resistance development, but by modulating tolerance. The identified RNA modification genes had not previously been associated with any antibiotic resistance phenotype. The fact that different tRNA modifications have opposite effects on tolerance to different antibiotics highlights the complexity of such a network, and shows that the observed phenotypes are not merely due to a general mistranslation effect. Since tRNA modifications affect codon decoding and accuracy, it is important to address how differential translation can generate proteome diversity, and eventually adaptation to antibiotics.

In particular, deletion of the *tgt* gene encoding tRNA-guanine transglycosylase (Tgt) in *V. cholerae* confers a strong growth defect in the presence of aminoglycosides at doses below the minimal inhibitory concentration (sub-MIC) (*Babosan et al., 2022*). Tgt incorporates queuosine (Q) in the place of guanosine (G) in the wobble position of four tRNAs with GUN anticodon (tRNA$^{Asp}_{GUC}$, tRNA$^{Asn}_{GUU}$, tRNA$^{Tyr}_{GUA}$, tRNA$^{His}_{GUG}$) (*Ehrenhofer-Murray, 2017*). The tRNAs with 'AUN' anticodons are not present in the genome, and thus each one of the four GUN tRNAs decodes two synonymous codons

(aspartate GAC/GAT, asparagine AAC/AAT, tyrosine TAC/TAT, histidine CAC/CAT which differ in the third position). Q34 is known to increase or decrease translation error rates in eukaryotes in a codon and organism-specific manner (*Ehrenhofer-Murray, 2017*; *Meier et al., 1985*). Q34 was shown to induce mild oxidative stress resistance in the eukaryotic parasite *Entamoeba histolytica*, the causative agent of amebic dysentery, and to attenuate its virulence (*Nagaraja et al., 2021*). In *E. coli*, the absence of Q34 modification was found to decrease mistranslation rates by tRNA$^{Tyr}$, while increasing it for tRNA$^{Asp}$ (*Manickam et al., 2016*; *Manickam et al., 2014*). No significant biological difference was found in *E. coli Δtgt* mutant, except for a slight defect in stationary phase viability (*Noguchi et al., 1982*), and more recently an involvement in biofilm formation (*Díaz Rullo and González Pastor, 2023*). Recent studies show that the *E. coli tgt* mutant is more sensitive to aminoglycosides but not to ampicillin nor spectinomycin and is more sensitive to oxidative stress but the molecular mechanisms were not elucidated (*Pollo-Oliveira et al., 2022*).

We asked here how queuosine (Q) modification by Tgt modulates the response to low doses of aminoglycosides. We find that *V. cholerae Δtgt* displays differential decoding of tyrosine TAC vs TAT codons. Molecular reporters, coupled to proteomics and in silico analysis, reveal that several proteins with codon usage biased toward TAT (vs TAC) are more efficiently translated in *Δtgt*. One of these proteins is RsxA, which prevents activation of SoxR oxidative stress response regulon (*Koo et al., 2003*). We propose that tobramycin treatment leads to increased expression of *tgt* and Q34 modification, which in turn allows for more efficient Sox regulon-related oxidative stress response, and better response to tobramycin. Lastly, bioinformatic analysis identified DNA repair gene transcripts with TAT codon bias as transcripts modulated by Q34 modification, which was confirmed by decreased UV susceptibility of *V. cholerae Δtgt*.

## Results

### Tobramycin tolerance is decreased in *Δtgt* without any difference in uptake

We performed competition experiments of *Δtgt* against the wild-type (WT) strain in the absence of stress and with various stresses including antibiotics (tobramycin, ciprofloxacin, carbenicillin) and oxidative agents (paraquat, $H_2O_2$). We confirmed *V. cholerae Δtgt* strain's growth defect in low-dose tobramycin (sub-MIC TOB) (*Figure 1A*) and that expression of *tgt* in trans restores growth in these conditions (*Figure 1B*). We further tested tolerance to lethal antibiotic concentrations by measuring survival after antibiotic treatment during 15 min to 4 hr with antibiotics at 5 times or 10 times the minimal inhibitory concentration. As expected, *Δtgt* is less tolerant than WT to tobramycin (*Figure 1C and D*), but had no impact in ciprofloxacin (CIP) or carbenicillin (CRB) (*Figure 1E and F*).

We asked whether the growth defect of *Δtgt* is due to increased aminoglycoside entry and/or a change in proton-motive force (PMF) (*Carvalho et al., 2021b*; *Lang et al., 2021*). We used a *ΔtolA* strain as a positive control for disruption of outer membrane integrity and aminoglycoside uptake (*Rivera et al., 1988*). No changes either in PMF (*Figure 1G*) or in uptake of the fluorescent aminoglycoside molecule Neomycin-Cy5 (*Figure 1H*; *Sabeti Azad et al., 2020*) were detected in the *Δtgt* strain, indicating that the increased susceptibility of *Δtgt* to TOB is not due to increased aminoglycoside entry into the *V. cholerae* cell.

### Overexpression of the canonical tRNA$^{Tyr}_{GUA}$ rescues growth of *Δtgt* in TOB

We next investigated whether all four tRNAs with GUN anticodon modified to QUN by Tgt are equally important for the TOB sensitivity phenotype of the *Δtgt* mutant: aspartate (Asp)/asparagine (Asn)/tyrosine (Tyr)/histidine (His). The absence of Q34 could have direct effects at the level of codon decoding but also indirect effects such as influencing tRNAs' degradation (*Kimura and Waldor, 2019*). qRT-PCR analysis of tRNA$^{Tyr}$ levels showed no major differences between WT and *Δtgt* strains, making it unlikely that the effect of Q34 modification on codon decoding is caused by altered synthesis or degradation of tRNA$^{Tyr}$ (*Figure 1—figure supplement 1A*). The levels of the other three tRNAs modified by Tgt also remained unchanged (*Figure 1—figure supplement 1A*). These results do not however exclude a more subtle or heterogeneous effect of Q34 modification on tRNA levels, which would be below the detection limits of the technique in a bacterial whole population.

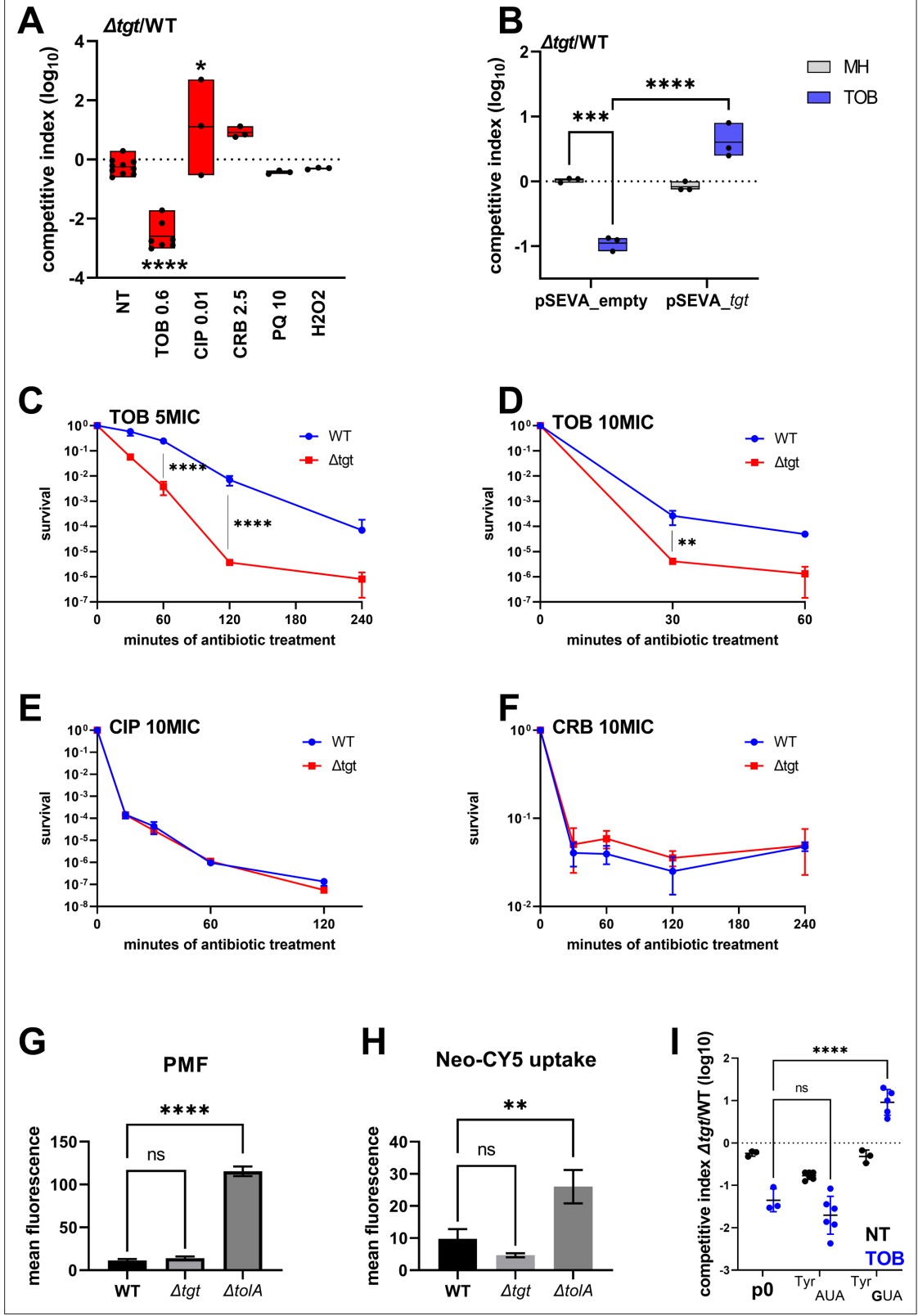

**Figure 1.** *V. cholerae Δtgt* shows decreased aminoglycoside tolerance. (**A**) Competition experiments between wild-type (WT) and *Δtgt* with the indicated antibiotic or oxidant at sub-MIC concentration. NT: non-treated; TOB: tobramycin 0.6 μg/mL; CIP: ciprofloxacin 0.01 μg/mL; CRB: carbenicillin 2.5 μg/mL; PQ: paraquat 10 μM; $H_2O_2$: 2 mM. (**B**) Competition experiments between WT and *Δtgt* carrying the indicated plasmids. pSEVA-*tgt*: plasmid expressing *tgt* using the XylS regulated Pm promoter, which is activated using sodium benzoate. MH: non-treated. (**C–F**) Survival of exponential phase

*Figure 1 continued on next page*

*Figure 1 continued*

cultures after various times of incubation (indicated in minutes on the X-axis) with the indicated antibiotic at lethal concentration: 5MIC: 5 times the MIC; 10MIC: 10 times the MIC. (**G**) Proton-motive force (PMF) measurement of exponential phase cultures using fluorescent MitoTracker dye, measured using flow cytometry. (**H**) Neomycin uptake measurement by flow cytometry using fluorescent Cy5 coupled neomycin. (**I**) Competition experiments between WT and *Δtgt* carrying either empty plasmid (p0), or a plasmid overexpressing tRNA$^{Tyr}$ with the native GUA anticodon, or with the synthetic AUA anticodon. The anticodon sequence is indicated (e.g. tRNA$^{Tyr}_{AUA}$ decodes the UAU codon). NT: non-treated; TOB: tobramycin 0.6 µg/mL. For multiple comparisons, we used one-way ANOVA. **** means $p<0.0001$, *** means $p<0.001$, ** means $p<0.01$, * means $p<0.05$. ns: non-significant. Number of replicates for each experiment: $3 < n < 8$.

The online version of this article includes the following figure supplement(s) for figure 1:

**Figure supplement 1.** Impact of tRNA overexpression on fitness during growth in sub-MIC tobramycin (TOB).

We next adopted a tRNA overexpression strategy from a high copy plasmid. The following tRNAs-$_{GUN}$ are the canonical tRNAs which are present in the genome: Tyr$_{GUA}$ (codon TAC), His$_{GUG}$ (codon CAC), two isoforms of Asn$_{GUU}$ (codon AAC), Asp$_{GUC}$ (codon GAT). The following tRNAs-$_{AUN}$ are synthetic tRNAs which are not present in the genome: Tyr$_{AUA}$, His$_{AUG}$, Asn$_{AUU}$, Asp$_{AUC}$. tRNA$^{Phe}_{GAA}$ was also used as non Tgt-modified control. Overexpression of tRNA$^{Tyr}_{GUA}$, but not tRNA$^{Tyr}_{AUA}$, rescues the *Δtgt* mutant's growth defect in sub-MIC TOB (**Figure 1I**). Overexpression of tRNA$^{His}_{AUG}$ also seemed to confer a benefit compared to empty plasmid (p0), but not as strong as tRNA$^{Tyr}_{GUA}$ (**Figure 1—figure supplement 1B**). We do not observe any major rescue of tobramycin-sensitive phenotypes when the other tRNAs are overexpressed, suggesting that changes in Tyr codon decoding is mostly responsible for the *Δtgt* mutant's tobramycin susceptibility phenotype.

## Q modification influences amino acid incorporation at tyrosine codons

We decided to measure the efficiency of amino acid incorporation at corresponding codons in *Δtgt*, using *gfp* reporters. First, we confirmed that GFP fluorescence from native GFP (encoded by *gfpmut3*) is not affected in *Δtgt* compared to WT (*gfp+* in **Figure 2**), indicating that there are no major differences on expression or folding of the GFP in *Δtgt*. We next constructed *gfp* fluorescent reporters by introducing within their coding sequence, stretches of repeated identical codons, for Asp/Asn/Tyr/His. This set of reporters revealed that the absence of Q34 leads to an increase of amino acid incorporation at Tyr TAT codons, both without and with tobramycin (**Figure 2A**, NT and TOB). This was not the case for Asp (**Figure 2B**) nor for Asn (**Figure 2D**), and we observed a slighter and more variable change for His (**Figure 2C**). No significant effect of *tgt* was observed for second near-cognate codons obtained by changing 1 base of the triplet for TAC and TAT codons (**Figure 2E**): Phe TTC/TTT, Cys TGT/TGC, Ser TCT/TCC (the third near-cognate stop codons TAA and TAG were not tested in this setup). Thus, Q34 modification strongly impacts the decoding of Tyr codons, and to a lesser extent His codons in this reporter system.

We also tested decoding reporters for TAT/TAC in WT and *Δtgt* overexpressing tRNA$^{Tyr}$ in trans (**Figure 1—figure supplement 1C**). The presence of the plasmid (empty p0) amplified differences between the two strains with decreased decoding of TAC (and increased TAT, as expected) in *Δtgt* compared to WT. Overexpression of tRNA$^{Tyr}_{GUA}$ did not significantly impact decoding of TAT and increased decoding of TAC, as expected. Since overexpression of tRNA$^{Tyr}_{GUA}$ rescues *Δtgt* in tobramycin (**Figure 1I**) and facilitates TAC decoding, this suggests that issues with TAC codon decoding contribute to the fitness defect observed in *Δtgt* upon growth with tobramycin. Overexpression of tRNA$^{Tyr}_{AUA}$ increased decoding of TAT in WT but did not change it in *Δtgt* where it is already high. Unexpectedly, overexpression of tRNA$^{Tyr}_{AUA}$ also increased decoding of TAC in WT. Thus, overexpression of tRNA$^{Tyr}_{AUA}$ possibly changes the equilibrium between the decoding of TAC vs TAT and may restore translation of TAC-enriched transcripts.

GFP reporters tested above with codon stretches were pivotal for the identification of codons for which decoding efficiency differs between WT and *Δtgt*, even though it's not a natural setup. We next developed a biologically relevant β-lactamase reporter tool to assess differences in the decoding of the tyrosine codons in WT and *Δtgt* strains. The amino acids Tyr103 and Asp129 of the β-lactamase were previously shown to be important for its function in resistance to β-lactam antibiotics, such as carbenicillin (*Doucet et al., 2004*; *Escobar et al., 1994*; *Jacob et al., 1990*).

We replaced the native Tyr103 TAC with the synonymous codon Tyr103 TAT (**Figure 3A**). While in the WT, both versions of β-lactamase conferred similar growth in carbenicillin with or without sub-MIC

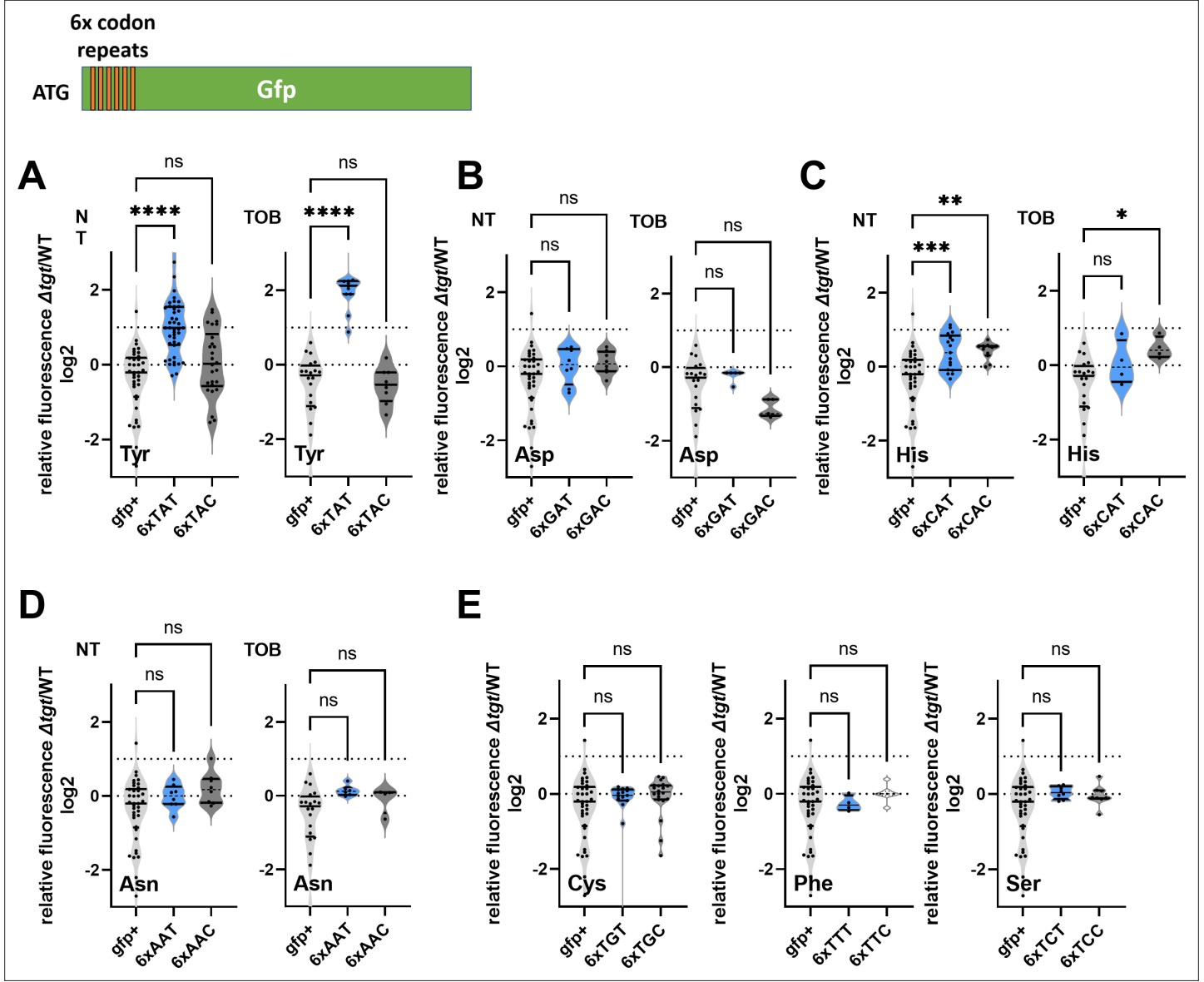

**Figure 2.** Codon decoding differences for *V. cholerae* wild-type (WT) and *Δtgt*. (**A–E**) Codon-specific translation efficiency in WT and *Δtgt* using 6xcodon stretches inserted in GFP. Y-axis represents the relative fluorescence of a given GFP in *Δtgt* over the same construct in WT. NT: non-treated; TOB: tobramycin at 0.4 µg/mL. Each specified codon is repeated 6× within the coding sequence of the GFP (e.g. TACTACTACTACTACTAC). For multiple comparisons, we used one-way ANOVA. **** means $p<0.0001$, *** means $p<0.001$, ** means $p<0.01$, * means $p<0.05$. ns: non-significant. Number of replicates for each experiment: 3<n, each dot represents one replicate gfp+ is the native gfp (gfpmut3) without any stretch.

TOB, in the *Δtgt* strain the Tyr-TAT version grows better than the Tyr TAC version upon exposure to TOB stress. This suggests a more efficient translation of the Tyr103 TAT β-lactamase mRNA, compared to the native Tyr103 TAC version, in stressed *Δtgt* strain.

Like Tyr103, Asp129 was shown to be important for resistance to β-lactams (***Doucet et al., 2004***; ***Escobar et al., 1994***; ***Jacob et al., 1990***). When we replaced the native Asp129 GAT with the synonymous codon Asp129 GAC, the GAC version did not appear to produce functional β-lactamase in *Δtgt* (***Figure 3B***), suggesting increased mistranslation or inefficient decoding of the GAC codon by tRNA$^{Asp}$ in the absence of Q. Decoding of GAT codon was also affected in *Δtgt* in the presence of tobramycin.

## Q modification impacts decoding fidelity in *V. cholerae*

To test whether a defect in Q34 modification influences the fidelity of translation in the presence and absence of tobramycin, previously developed reporter tools were used (***Fabret and Namy, 2021***), to

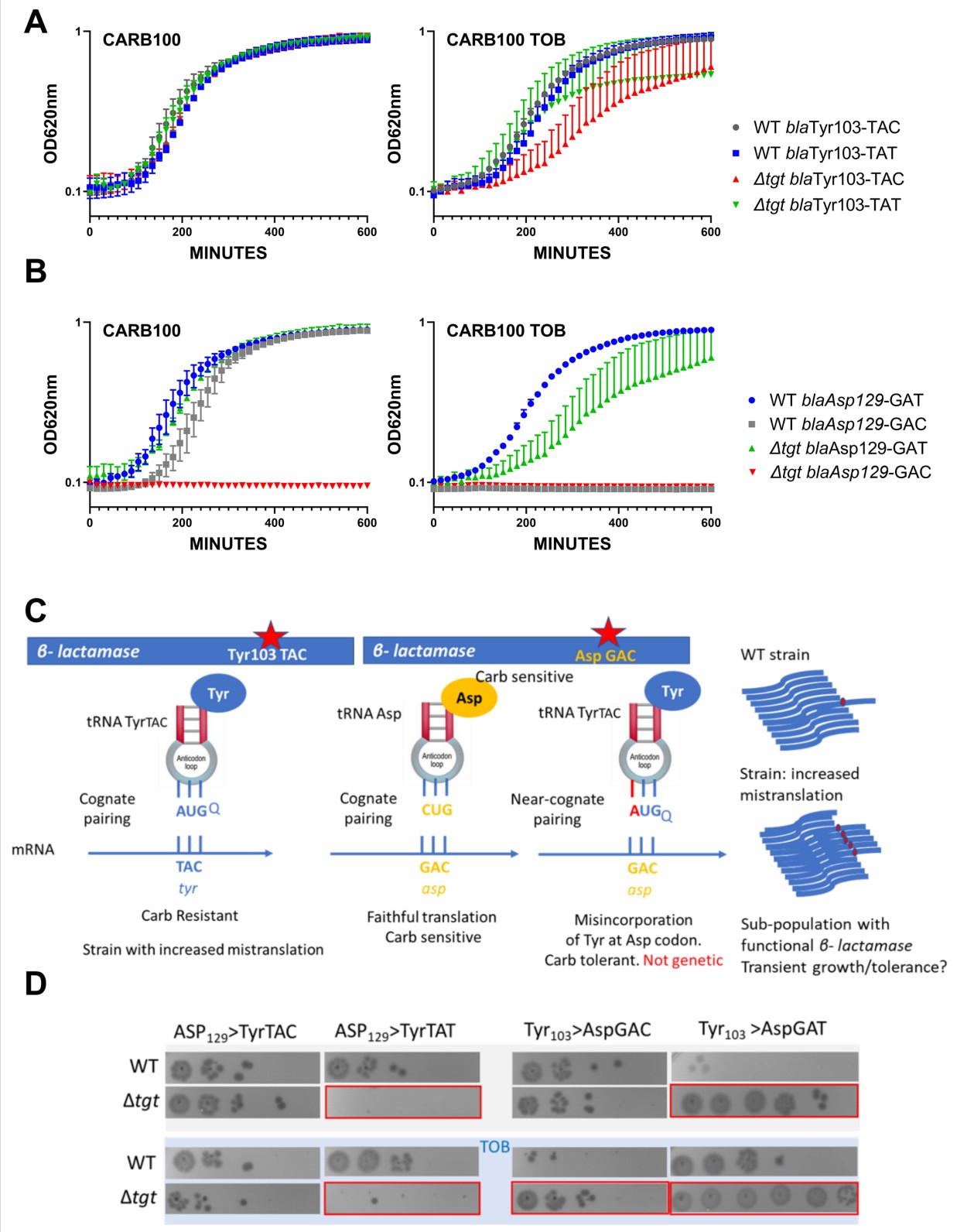

**Figure 3.** Differential translation at tyrosine and aspartate codons evaluated by growth on carbenicillin of mutated β-lactamase reporters. (**A**) The catalytic tyrosine at position 103 was tested in its native sequence (TAC), or with the synonymous TAT mutation. (**B**) The catalytic aspartate at position 129 was tested in its native sequence (GAT), or with the synonymous GAC mutation. (**A and B**). Growth on microtiter plates. Growth was followed by measuring the OD 620 nm every 15 min during 600 min. CARB was used at 100 µg/mL. When present, TOB was used at 0.2 µg/mL (20% of MIC). Curves

*Figure 3 continued on next page*

*Figure 3 continued*

represent mean and standard errors from biological duplicates. (**C**) Rationale. (**D**) Cells were grown to early exponential phase without carbenicillin, and with (down) or without (top) tobramycin 20% of MIC, and treated with carbenicillin at 10× MIC for 20 hr. Dilutions were spotted on plate without carbenicillin. Surviving cells shown here are sensitive to carbenicillin (no growth on carbenicillin containing plates), suggesting that increased or decreased survival was due to increased (erroneous translation) or decreased (faithful translation) β-lactamase activity at the time of treatment. Number of replicates for each experiment: 3. One representative experiment is shown.

The online version of this article includes the following figure supplement(s) for figure 3:

**Figure supplement 1.** Q modification impacts decoding fidelity in *V. cholerae*.

measure stop codons readthrough in *V. cholerae Δtgt* and WT strains. The system consists of vectors containing readthrough promoting signals inserted between the *lacZ* and *luc* sequences, encoding β-galactosidase and luciferase, respectively. Luciferase activity reflects the readthrough efficiency, while β-galactosidase activity serves as an internal control of expression level, integrating a number of possible sources of variability (plasmid copy number, transcriptional activity, mRNA stability, and translation rate). We found increased readthrough at stop codons UAA and to a lesser extent at UAG for *Δtgt*, and this increase was amplified for UAG in presence of tobramycin (*Figure 3—figure supplement 1*, stop readthrough). In the case of UAA, tobramycin appears to decrease readthrough, this may be artifactual, due to the toxic effect of tobramycin on *Δtgt*.

Mistranslation at specific codons can also impact protein synthesis. To further investigate mistranslation levels by tRNA$^{Tyr}$ in WT and *Δtgt*, we designed a set of *gfp* mutants where the codon for the catalytic tyrosine required for fluorescence (TAT at position 66) was substituted by near-cognate codons (*Figure 3—figure supplement 1*). Results suggest that in this sequence context, particularly in the presence of tobramycin, non-modified tRNA$^{Tyr}$ mistakenly decodes Asp GAC, His CAC, and also Ser UCC, Ala GCU, Gly GGU, Leu CUU, and Val GUC codons, suggesting that Q34 increases the fidelity of tRNA$^{Tyr}$.

In parallel, we replaced Tyr103 of the β-lactamase described above, with Asp codons GAT or GAC. The expression of the resulting mutant β-lactamase is expected to yield a carbenicillin-sensitive phenotype. In this system, increased tyrosine misincorporation (more mistakes) by tRNA$^{Tyr}$ at the mutated Asp codon will lead to increased synthesis of active β-lactamase, which can be evaluated by carbenicillin tolerance tests. As such, amino acid misincorporation leads here to phenotypic (transient) tolerance, while genetic reversion mutations result in resistance (growth on carbenicillin). The rationale is summarized in *Figure 3C*. When the Tyr103 codon was replaced with either Asp codons, we observe increased β-lactamase tolerance (*Figure 3D*, left), suggesting increased misincorporation of tyrosine by tRNA$^{Tyr}$ at Asp codons in the absence of Q, again suggesting that Q34 prevents misdecoding of Asp codons by tRNA$^{Tyr}$.

In order to test any effect on an additional tRNA modified by Tgt, namely tRNA$^{Asp}$, we mutated the Asp129 (GAT) codon of the β-lactamase. When Asp129 was mutated to Tyr TAT (*Figure 3D*, right), we observe reduced tolerance in *Δtgt*, but not when it was mutated to Tyr TAC, suggesting less misincorporation of aspartate by tRNA$^{Asp}$ at the Tyr UAU codon in the absence of Q. In summary, absence of Q34 increases misdecoding by tRNA$^{Tyr}$ at Asp codons, but decreases misdecoding by tRNA$^{Asp}$ at Tyr UAU.

This supports the fact that tRNA Q34 modification is involved in translation fidelity during antibiotic stress, and that the effects can be different on different tRNAs, e.g., tRNA$^{Tyr}$ and tRNA$^{Asp}$ tested here.

## Proteomics study identifies RsxA among factors for which translation is most impacted in *Δtgt*

These observations show a link between Q34 modification of tRNA, differential decoding of Tyr codons (among others), and susceptibility to aminoglycosides. We hypothesized that proteins that are differentially translated according to their Tyr codon usage could be involved in the decreased efficiency of the response to aminoglycoside stress in *Δtgt*. We conducted a proteomics study comparing WT vs *Δtgt*, in the absence and presence of tobramycin (proteomics *Supplementary file 1* and *Figure 4*). Loss of Q34 results in generally decreased detection of many proteins in tobramycin (shift toward the left in the volcano plot; *Figure 4AB*), and in increases in the levels of 96 proteins. Among those, RsxA (encoded by VC1017) is 13-fold more abundant in the *Δtgt* strain compared to WT in tobramycin. RsxA is part of an anti-SoxR complex. SoxR is an oxidative stress response regulator (*Koo et al., 2003*)

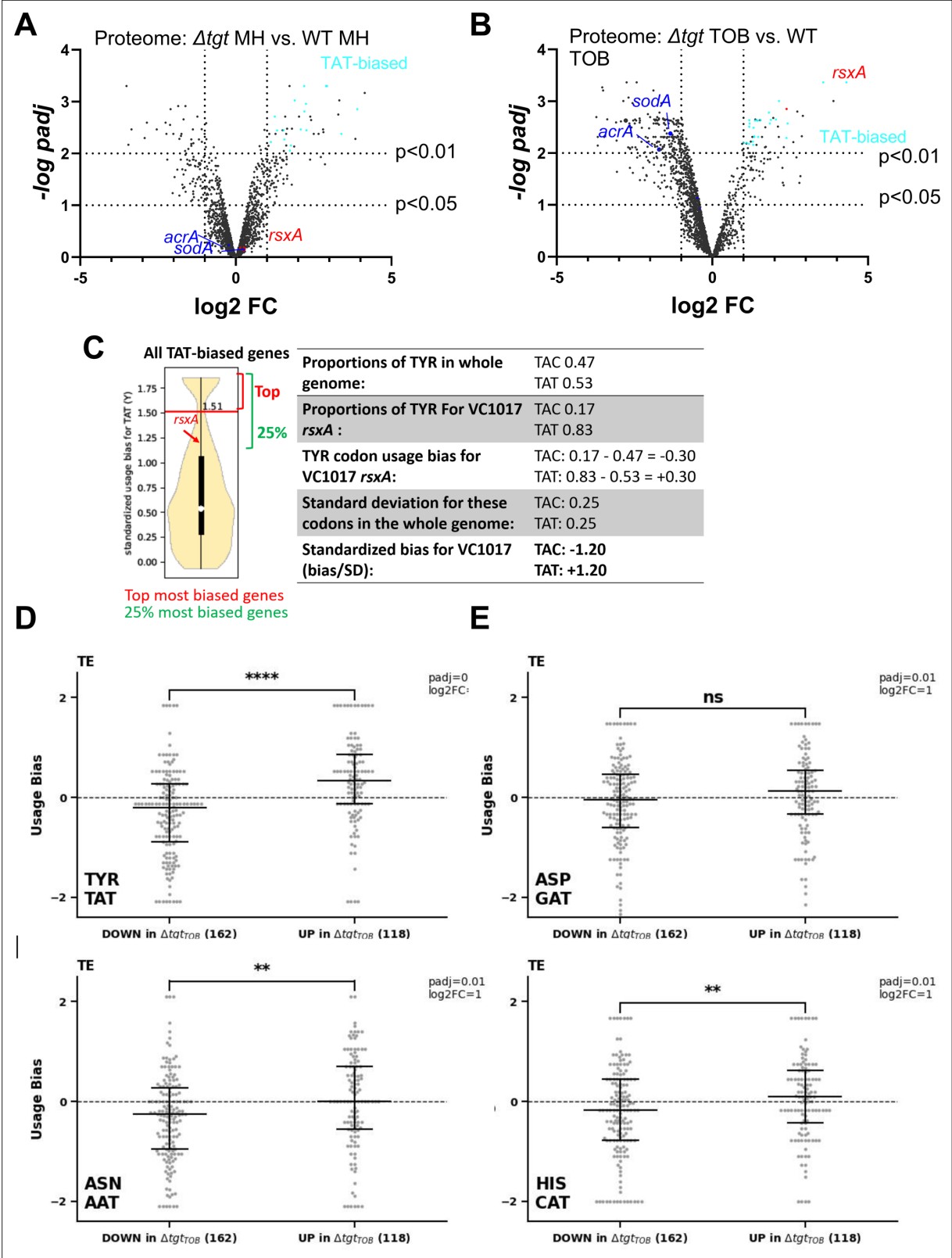

**Figure 4.** Post-transcriptional regulation in *Δtgt* and Tyr codon usage bias. (**A and B**) Volcano plots showing less and more abundant proteins in proteomics analysis (performed in triplicates) in *Δtgt* compared to wild-type (WT) during growth without antibiotics (**A**, where MH is the growth medium), or in sub-MIC TOB at 0.4 µg/mL (**B**). (**C**) Codon usage bias calculation example with VC1017 *rsxA* tyrosine TAC and TAT codons. (**D–G**) Plots showing codon usage bias for genes lists that are up or down in ribosome profiling analysis (performed in triplicates), normalized to RNA-seq values for

*Figure 4 continued on next page*

*Figure 4 continued*

each gene, for codons decoded by tRNAs with Q modification. Each dot represents one gene of the lists TE stands for translation efficiency, i.e. Riboseq data normalized to total transcriptome.

that controls *sodA* (VC2694, superoxide dismutase) and *acrA* (VC0913, efflux), among other genes of the regulon. The Rsx complex reduces and inactivates SoxR, preventing the induction of the regulon. Consistently, we find that the levels of SodA and AcrA proteins are decreased in *Δtgt* compared to WT in TOB (indicated in *Figure 4B*).

With 83% of Tyr TAT codons, instead of the expected 53% average, RsxA has a clear codon usage bias. To test whether some of the differentially abundant protein groups in the Q34-deficient mutant show similar biases, the Tyr codon usage was calculated for the 96 more abundant and 195 less abundant proteins expressed in TOB. More abundant proteins in *Δtgt* TOB with a codon usage bias toward TAT vs TAC are represented as light blue dots in *Figure 4A and B*. No statistically significant difference was detected for TAT codon usage in neither sets of proteins. Thus, one cannot draw conclusions or infer predictions about codon decoding efficiencies in a tRNA modification mutant such as *Δtgt* from the proteomics data alone.

We thus performed Ribo-seq (ribosome profiling) analysis on extracts from WT and *Δtgt* strains grown in the presence of sub-MIC TOB. Unlike for eukaryotes, technical limitations (e.g. the RNase which is used to display significant sequence specificity) do not necessarily allow to obtain codon resolution in bacteria (*Mohammad et al., 2019*). However, we determined 159 transcripts with increased and 197 with decreased translation in *Δtgt* and we plotted their standardized codon usage bias for codons of interest (*Figure 4D–G*). The calculation of this value is explained in details in the Materials and methods section, and shown for *rsxA* as example in *Figure 4C*. Briefly, we took as reference the mean proportion of the codons of interest in the genome (e.g. for tyrosine: TAT = 0.53 and TAC = 0.47, meaning that for a random *V. cholerae* gene, 53% of tyrosine codons are TAT). For each gene, we calculated the proportion of each codon (e.g. for *rsxA*: TAT = 0.83 and TAC = 0.17). We next calculated the codon usage bias as the difference between a given gene's codon usage and the mean codon usage (e.g. for *rsxA*, the codon usage bias for TAT is 0.83–0.53=+0.30). Finally, in order to consider the codon distribution on the genome and obtain statistically significant values, we calculated standardized bias by dividing the codon usage bias by standard deviation for each codon (e.g. for *rsxA*, 0.30/0.25 = +1.20). This is done to adjust standard deviation to 1, and thus to get comparable (standardized) values, for each codon.

Ribo-seq data (*Supplementary file 2*) shows that TAT codon usage was decreased in the list of transcripts with decreased translation efficiency in *Δtgt*, while it was increased in the list of transcripts with increased translation efficiency (*Figure 4D and E*). No difference was detected for AAT and CAT (*Figure 4F and G*).

In addition to mistranslation and codon decoding efficiency, other factors also influence detected protein levels, such as transcription, degradation, etc. Moreover, the localization and sequence context of the codons for which the efficiency of translation is impacted may be important. Nevertheless, as translation of proteins with a codon usage bias toward TAC or TAT may be impacted in *Δtgt*, and as the most abundant protein RsxA in *Δtgt* in TOB shows a strong TAT bias, we decided to evaluate whether RsxA is post-transcriptionally regulated by the Q34 modification and whether it may affect fitness in the presence of tobramycin.

## RsxA is post-transcriptionally upregulated in *Δtgt* due to more efficient decoding of tyrosine TAT codons in the absence of Q34 modification

Transcriptomic analysis comparing at least twofold differentially expressed genes between *V. cholerae Δtgt* and WT strains (*Supplementary file 3*) showed that, respectively, 53 and 26 genes were significantly downregulated in MH and tobramycin, and 34 were up in tobramycin. Gene ontology (GO) enrichment analysis showed that the most impacted GO categories were bacteriocin transport and iron import into the cell (45- and 40-fold enriched) in MH, and proteolysis and response to heat (38- and 15-fold enriched) in TOB. In both conditions, the levels of *rsxA* transcript remained unchanged.

RsxA carries six tyrosine codons among which the first five are TAT and the last one is TAC. RsxA is 13-fold more abundant in *Δtgt* than WT, but transcript levels measured by digital RT-PCR are comparable in both strains (*Figure 5A*), consistent with RNA-seq data. We constructed transcriptional

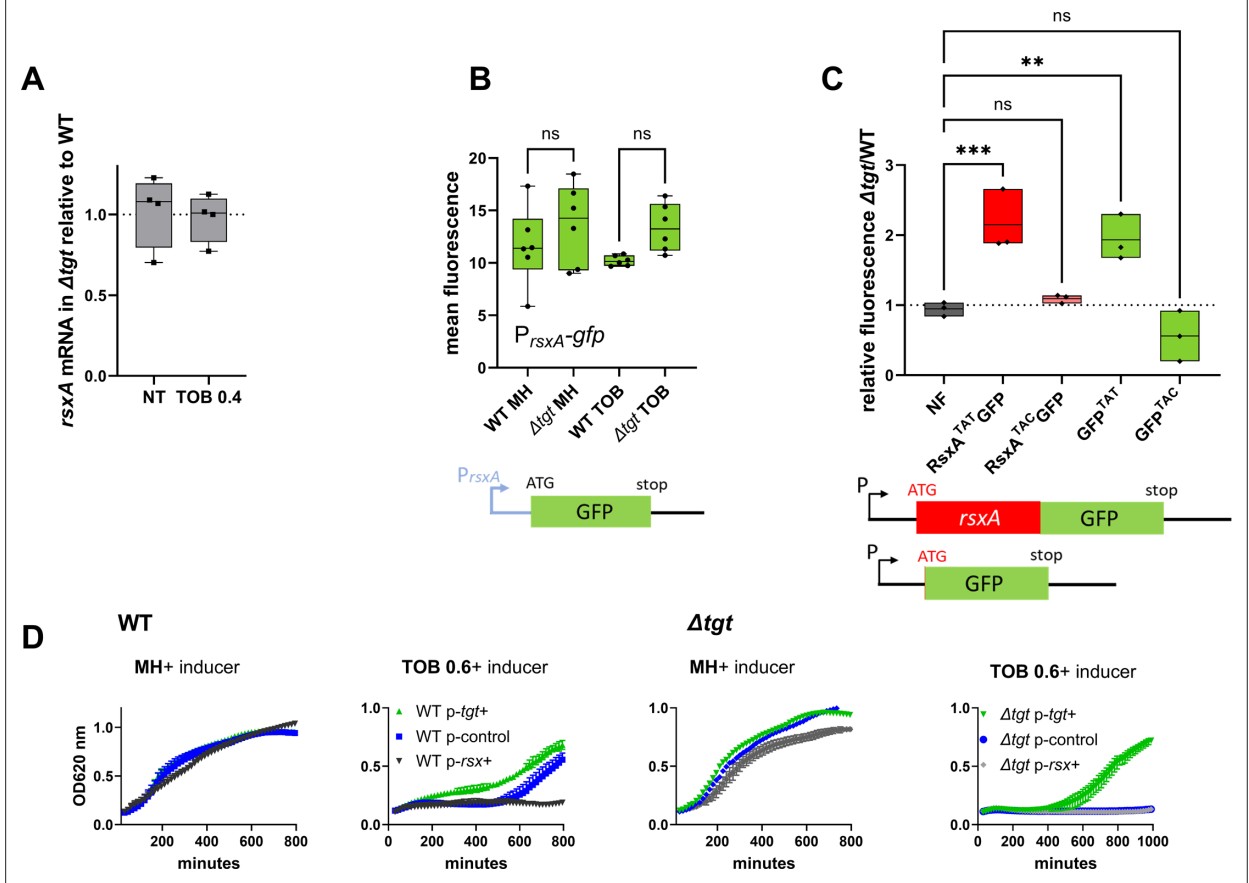

**Figure 5.** Post-transcriptional upregulation of RsxA in *Δtgt* due to a Tyr codon bias toward TAT and toxicity in sub-MIC TOB. (**A**) *rsxA* mRNA levels measured by digital RT-PCR. 4 biological replicates. (**B**) Transcriptional expression levels from the *rsxA* promoter measured by flow cytometry. 6 biological replicates. (**C**) Translational fusion of *rsxA* to *gfp* and *gfp* alone, with differences in codon usage. 3 biological replicates. One-way ANOVA was used to determine the statistical differences (p-value) between groups. *** means p<0.001, ** means p<0.01, ns: non-significant. (**D**) Growth curves in microtiter plate reader, in *V. cholerae* in indicated conditions. P-tgt+: pSEVA expressing *tgt*. P-rsx+: pSEVA expressing rsxA. P-control: empty pSEVA. Inducer: DAPG (see Materials and methods). 2 biological replicates and standard errors are shown for each curve.

The online version of this article includes the following figure supplement(s) for figure 5:

**Figure supplement 1.** Post-transcriptional upregulation of RsxA in *Δtgt* due to a Tyr codon bias toward TAT and toxicity in sub-MIC TOB.

**Figure supplement 2.** rsxA levels impact growth in the presence of sub-MIC TOB in *E. coli*.

and translational *gfp* fusions in order to evaluate the expression of *rsxA* in WT and *Δtgt* strains. As expected from digital RT-PCR results, no significant differences in fluorescence were observed for the transcriptional fusion of the *rsxA* promoter with *gfp* (**Figure 5B**), excluding transcriptional regulation of *rsxA* in this context. For translational fusions, we used either the native *rsxA* sequence bearing five TAT+1 TAC codons, or a mutant *rsxA* allele carrying all six TAC codons (hereafter called respectively RsxA^TAT and RsxA^TAC). Confirming the proteomics results, the RsxA^TAT-GFP fusion was more fluorescent in the *Δtgt* mutant, but not the RsxA^TAC-GFP one (**Figure 5C** and detailed flow cytometry data in **Figure 5—figure supplement 1A–C**). Since increased *rsxA* expression appeared to be somewhat toxic for growth, and in order to test translation on a reporter which confers no growth defect, we chose to test directly the translation of *gfp*, which originally carries four TAT (36%) and seven TAC (64%) codons in its native sequence. We constructed two synonymous versions of the GFP protein, with all 11 tyrosine codons either changed to TAT or to TAC. Similar to what we observed with *rsxA*, the GFP^TAT version, but not the GFP^TAC one, generated more fluorescence in the *Δtgt* background, (**Figure 5C** and detailed flow cytometry data in **Figure 5—figure supplement 1D–F**).

Since not all TAT-biased proteins are found to be enriched in *Δtgt* proteomics data, the sequence context surrounding TAT codons could affect their decoding. To illustrate this, we inserted after

the *gfp* start codon, various tyrosine containing sequences displayed by *rsxA* (*Figure 5—figure supplement 1G*). The native tyrosines were all TAT codons, our synthetic constructs were either TAT or TAC, while keeping the remaining sequence unchanged. We observe that the production of GFP carrying the TEY$^{TAT}$LLL sequence from RsxA is increased in *Δtgt* compared to WT, while it is unchanged with TEY$^{TAC}$LLL. However, production of the GFP with the sequences LY$^{TAT}$RLL/ LY$^{TAC}$RLL and EY$^{TAT}$LR/ EY$^{TAC}$LR was unaffected (or even decreased for the latter) by the absence of *tgt*. Overall, our results demonstrate that RsxA is upregulated in the *Δtgt* strain at the translational level, and that proteins with a codon usage bias toward tyrosine TAT are prone to be more efficiently translated in the absence of Q modification, but this is also dependent on the sequence context.

## Increased expression of RsxA hampers growth in sub-MIC TOB

We asked whether high levels of RsxA could be responsible for *Δtgt* strain's increased sensitivity to tobramycin. *rsxA* cannot be deleted since it is essential in *V. cholerae* (see our TN-seq data; *Babosan et al., 2022*; *Negro et al., 2019*). We overexpressed *rsxA* from an inducible plasmid in WT strain. In the presence of tobramycin, overexpression of *rsxA* in the WT strain strongly reduces growth (*Figure 5D* with inducer, black curve compared to blue), while overexpression of *tgt* restores growth of the *Δtgt* strain (*Figure 5D* with inducer, green curve). This shows that increased *rsxA* levels can be toxic during growth in sub-MIC TOB and is consistent with decreased growth of the *Δtgt* strain.

Unlike for *V. cholerae*, *rsxA* is not an essential gene in *E. coli*, and does not bear a TAT bias. It has however the same function. Note that this is not the first instance where we observe differences between *E. coli* and *V. cholerae* regarding oxidative stress (*Baharoglu et al., 2013*; *Baharoglu and Mazel, 2011*; *Baharoglu and Mazel, 2014*) and respiration processes (*Krin et al., 2023*): the dispensability of *rsxA* in *E. coli* could either be due to the presence of an additional redundant pathway in this species, or alternatively to general differences in how the two species respond to stress.

In order to confirm that the presence of RsxA can be toxic during growth in tobramycin, we additionally performed competition experiments in *E. coli* with simple and double mutants of *tgt* and *rsxA*. Since *Δtgt* strain's growth is more affected than WT at TOB 0.5 μg/mL (indicated with an arrow in *Figure 5—figure supplement 2A*), we chose this concentration for competition and growth experiments. The results confirm that inactivation of *rsxA* in *Δtgt* restores fitness in tobramycin (*Figure 5—figure supplement 2B*), and that overproduction of RsxA decreases growth in TOB. *tgt* transcription is repressed by CRP and slightly induced by tobramycin in *V. cholerae tgt* was previously observed to be upregulated in *E. coli* isolates from urinary tract infection (*Bielecki et al., 2014*) and in *V. cholerae* after mitomycin C treatment (through indirect SOS induction; *Krin et al., 2018*). We measured *tgt* transcript levels using digital RT-PCR in various transcriptional regulator deficient mutants (iron uptake repressor Fur, general stress response and stationary phase sigma factor RpoS and carbon catabolite control regulator CRP), as well as upon exposure to antibiotics, particularly because *tgt* is required for growth in sub-MIC TOB. We also tested the iron chelator dipyridyl (DP), the oxidant agent paraquat (PQ), and serine hydroxamate (SHX) which induces the stringent response.

Among all tested conditions, we found that sub-MIC tobramycin and the stringent response mildly increase *tgt* transcript levels, while the carbon catabolite regulator CRP appears to repress it (*Figure 6A*). We found a sequence between ATG −129 and −114: TTC**G**C$^{AGGGAA}$A**C**GCG which shows some similarity (in blue) to the *V. cholerae* CRP binding consensus (T/A)$_1$(G/T)$_2$(T/C)$_3$**G**$_4$(A/C)$_5^{NNNNN-}$ $^N$(T/G)$_{12}$**C**$_{13}$(A/G)$_{14}$(C/A)$_{15}$(T/A)$_{16}$. However, CRP binding was not previously detected by ChIP-seq in the promoter region of *tgt* in *V. cholerae* (*Manneh-Roussel et al., 2018*). CRP binding could be transitory or the repression of *tgt* expression by CRP could be an indirect effect.

Regarding induction by tobramycin, the mechanism remains to be determined. We previously showed that sub-MIC TOB induces the stringent response (*Babosan et al., 2022*; *Carvalho et al., 2021a*). Since induction of *tgt* expression by SHX and by tobramycin seems to be in the same order of magnitude, we hypothesized that tobramycin could induce *tgt* through the activation of the stringent response. Using a *P1rrnB-gfp* fusion (*Babosan et al., 2022*), which is downregulated upon stringent response induction (*Kolmsee et al., 2011*; *Figure 6B*), we found that the stringent response is significantly induced by tobramycin, both in WT and *Δtgt*. Tobramycin may induce *tgt* expression through stringent response activation or through an independent pathway.

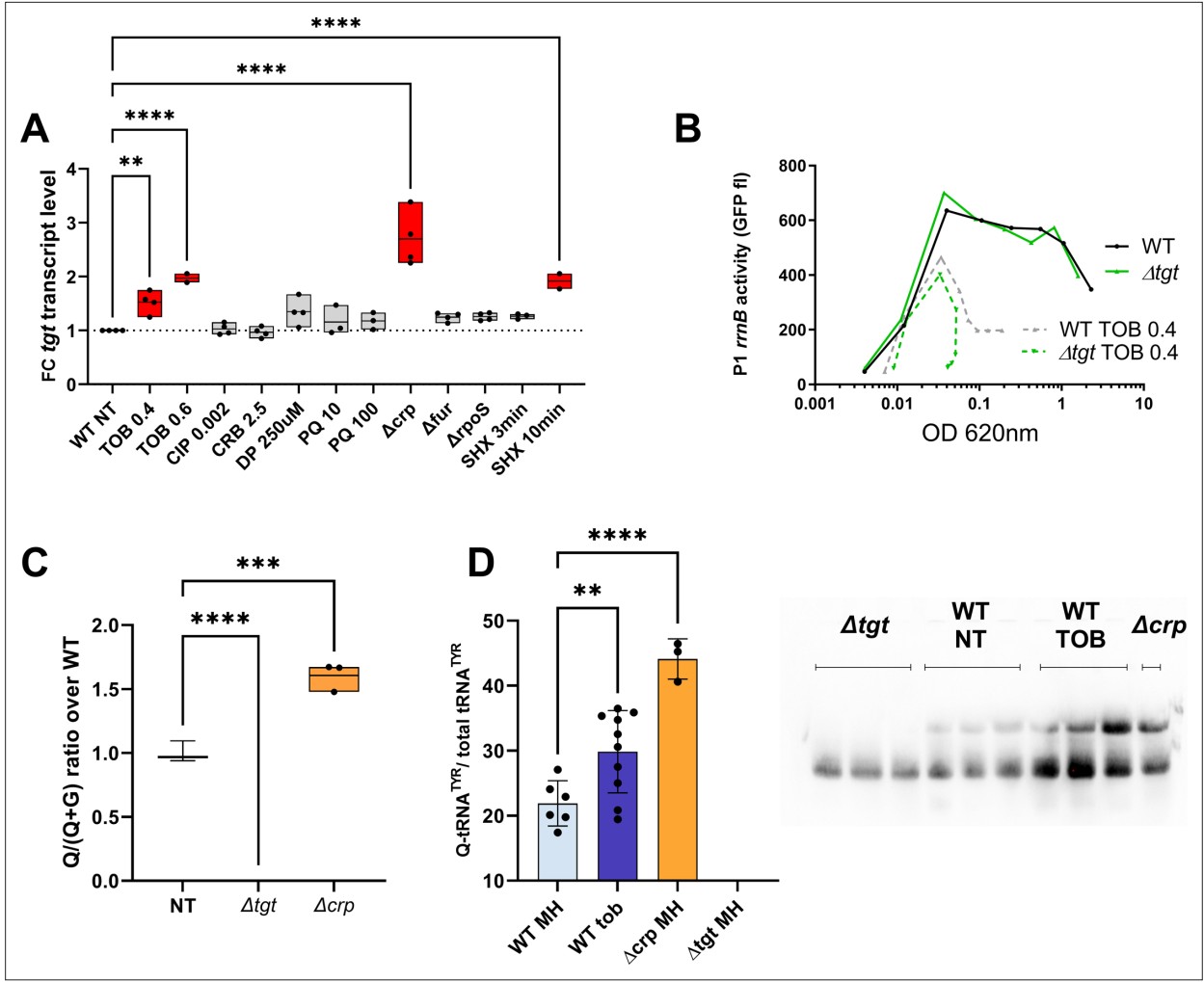

**Figure 6.** Regulation of *tgt* expression and tRNA Q levels. (**A**) *tgt* transcript levels measured by digital-RT-PCR. Y-axis represents relative abundance compared to the non-treated (NT) condition in wild-type (WT). (**B**) Stringent response induction measured with *P1rrn-gfp* reporter shown as fluorescence (Y-axis) as a function of growth (X-axis: OD 600 nm). (**C**) Q levels in tRNA-enriched RNA extracts, measured by mass spectrometry. (**D**) Northern blot and quantification of Q levels in tRNA$^{Tyr}$. The lower band in the gel corresponds to unmodified tRNA$^{Tyr}$. The upper band corresponds to Q-modified tRNA-Tyr. *Δtgt* is the negative control without modification of tRNA$^{Tyr}$. A representative gel is shown. Histograms show the quantification of Q-modified tRNA$^{Tyr}$ over total tRNA$^{Tyr}$, as follows: Q-modified tRNA$^{Tyr}$/(Q-modified tRNA$^{Tyr}$+tRNA$^{Tyr}$ without Q)=upper band/(upper band+lower band). 2.5 µg in tRNA-enriched RNA extracts were deposited in lanes WT NT, WT TOB, and *Δcrp*. 0.9 µg was deposited in lanes *Δtgt*. Number of replicates for each experiment: 3. For multiple comparisons, we used one-way ANOVA (for **A, C, E**). **** means p<0.0001, *** means p<0.001, ** means p<0.01. Only significant differences are shown.

The online version of this article includes the following source data and figure supplement(s) for figure 6:

**Source data 1.** PDF file containing the original northern blot for **Figure 6D**, indicating the relevant bands and treatments.

**Source data 2.** Original files for northern blot analysis displayed in **Figure 6D**.

**Figure supplement 1.** Q detection and quantification by IO4⁻ oxidation coupled to deep sequencing.

## Q modification levels can be dynamic and are directly influenced by *tgt* transcription levels

We have identified conditions regulating *tgt* expression. We next addressed whether up/downregulation of *tgt* affects the actual Q34 modification levels of tRNA. We measured Q34 levels by mass spectrometry in WT and the *Δcrp* strain, where the strongest impact on *tgt* expression was observed (**Figure 6C**). We find a significant 1.6-fold increase in Q34 levels in *Δcrp*. We also tested the effect of sub-MIC TOB, but smaller differences are probably not detected using our approach of mass spectrometry in bulk cultures.

In order to get deeper insight into modification level of *V. cholerae* tRNAs potentially having Q34 modification, we decided to adapt a recently published protocol for the detection and quantification of queuosine by deep sequencing (*Katanski et al., 2022*). This allowed us to validate the presence of Q34 modification in the *V. cholerae* tRNAs Asp, His, and Asn GTT2 and precisely measure its level and modulation under different growth conditions (*Figure 6—figure supplement 1*). We also showed that Q34 detection is robust and reproducible, and reveals increased Q34 content for tRNA^His and tRNA^Asn in Δ*crp* strain where *tgt* expression was induced, while for tRNA^Asp Q34 level remains relatively constant. *V. cholerae* tRNA^AsnGTT1 is very low expressed and likely contains only sub-stoichiometric amounts of Q34, while analysis of tRNA^Tyr is impeded by the presence of other modifications in the anticodon loop (namely i^6A37 or its derivatives), which prevents the correct mapping and quantification of Q34 modifications using deletion signature.

In order to evaluate Q34 levels in tRNA^Tyr more specifically, we performed *N*-acryloyl-3-aminophenylboronic acid (APB) northern blots allowing visualization and quantification of Q-modified and unmodified tRNAs (*Cirzi and Tuorto, 2021*). As anticipated from increased *tgt* expression, Q-modified tRNA levels were strongly increased in Δ*crp* strain. Sub-MIC TOB also increases the proportion of Q34 containing tRNA^Tyr compared to the non-treated condition (*Figure 6D*). However, this result was variable suggesting a subtle fine-tuning of the Q34 levels depending on growth state (optical density) and TOB concentration. These results show that tRNA Q34 modification levels are dynamic and correlate with variations in *tgt* expression, depending on the tRNA.

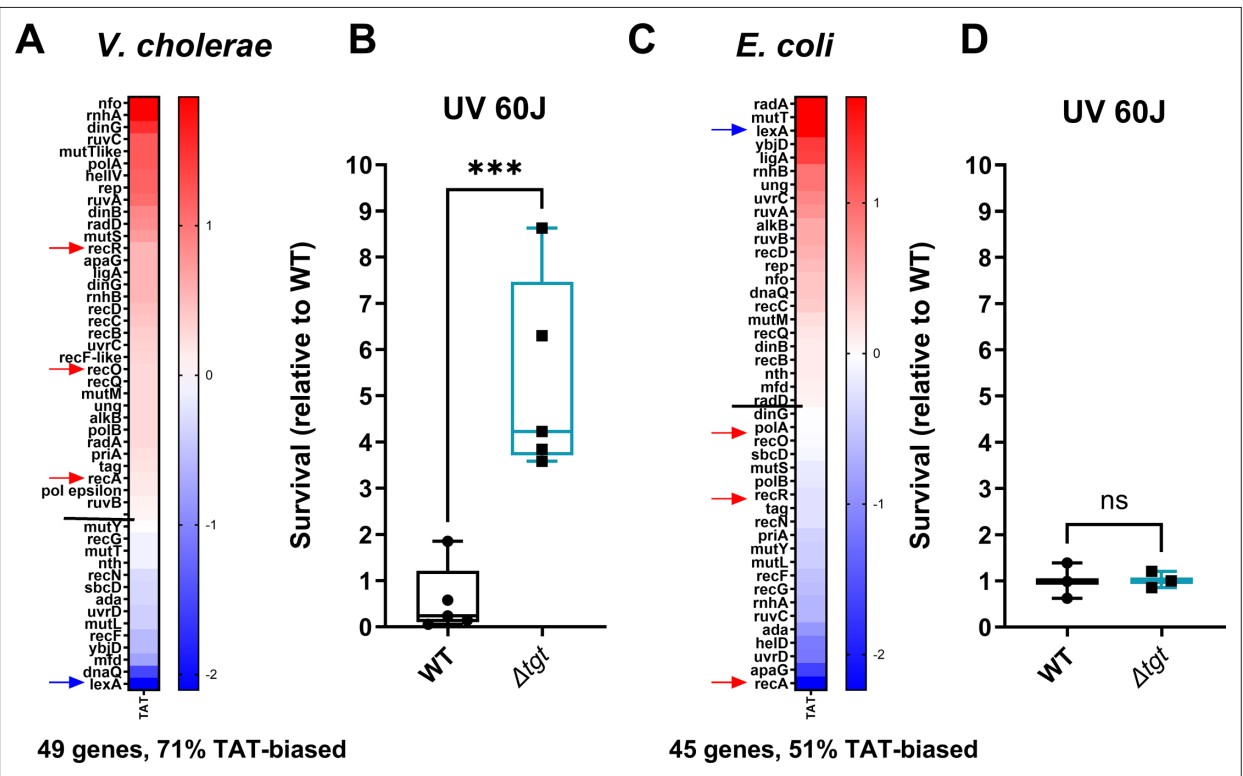

**Figure 7.** DNA repair after UV irradiation is more efficient in *V. cholerae* Δ*tgt*. (**A and C**) Tyrosine codon usage of DNA repair genes A: in *V. cholerae*. C: in *E. coli*. Red indicates positive codon usage bias, i.e., TAT bias. Blue indicates negative codon usage bias for TAT, i.e., TAC bias. (**B and D**) Survival of Δ*tgt* relative to wild-type (WT) after UV irradiation (linear scale) B: in *V. cholerae*. (**D**) in *E. coli*. For multiple comparisons, we used one-way ANOVA. **** means p<0.0001, ns: non-significant.

The online version of this article includes the following figure supplement(s) for figure 7:

**Figure supplement 1.** Codon usage.

## DNA repair genes are TAT-biased

We further analyzed in silico the codon usage of *V. cholerae* genome, and for each gene, we assigned a codon usage value to each codon (*Figure 4C* and doi:10.5281/zenodo.6875293). This allowed the generation of lists of genes with divergent codon usage, for each codon.

For genes with a tyrosine codon usage bias toward TAT in *V. cholerae*, GO enrichment analysis (*Figure 7—figure supplement 1*) highlights the DNA repair category with a p-value of $2.28×10^{-2}$ (*Figure 7—figure supplement 1C*). *Figure 7A* shows Tyr codon usage of *V. cholerae* DNA repair genes. We hypothesized that translation of DNA repair transcripts could be more efficient in *Δtgt*, and that such basal pre-induction would be beneficial during genotoxic treatments as UV irradiation (single-stranded DNA breaks). UV-associated DNA damage is repaired through RecA-, RecFOR-, and RuvAB-dependent homologous recombination. Five of these genes, *recO*, *recR*, *recA*, *ruvA*, and *ruvB*, are biased toward TAT in *V. cholerae* (*Figure 7A*, red arrows), while their repressor LexA bears a strong bias toward TAC. DNA repair genes (e.g. *ruvA* with 80% TAT, *ruvB* with 83% TAT, *dinB* with 75% TAT) were also found to be up for *Δtgt* in the Ribo-seq data, with unchanged transcription levels. *V. cholerae Δtgt* appears to be four to nine times more resistant to UV irradiation than the WT strain (*Figure 7B*). Better response to UV in the *V. cholerae Δtgt* strain is consistent with the possibility of

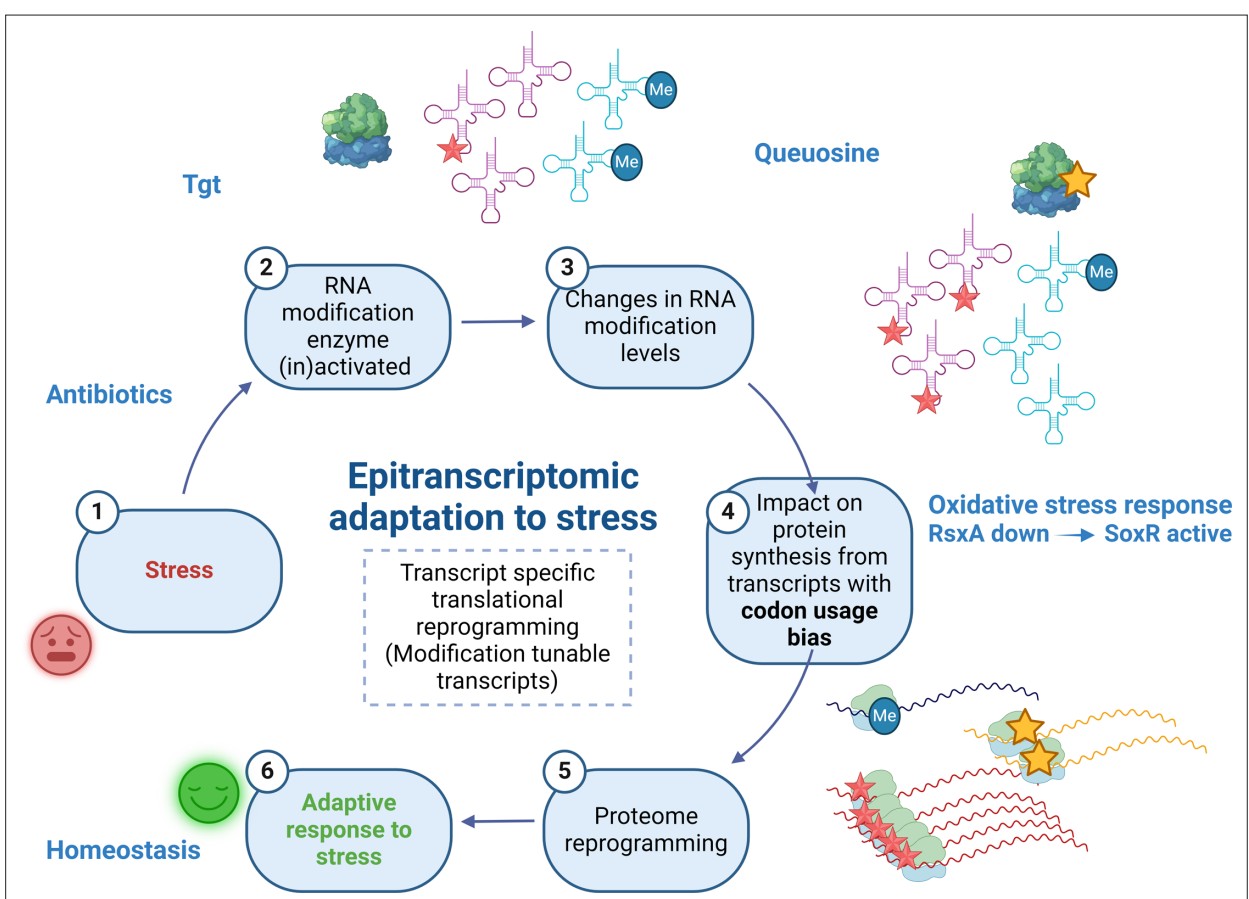

**Figure 8.** Model. Upon exposure to sub-MIC aminoglycosides, the expression of *tgt* is upregulated in *V. cholerae* and influences the decoding of tyrosine TAC vs TAT codons. This leads to differential translation from transcripts bearing a codon usage bias for tyrosine codons. The *rsxA* transcript bears a tyrosine codon bias and its translation can be tuned by tRNA Q modification. RsxA is an anti-SoxR factor. SoxR controls a regulon involved in oxidative stress response. When RsxA levels are high, SoxR is increasingly inactivated and oxidative stress response efficiency decreases. It has been previously shown that sub-MIC aminoglycosides trigger oxidative stress in *V. cholerae* (*Baharoglu et al., 2013*). Increasing RsxA levels thus reduce fitness in TOB by hampering efficient oxidative stress response. As a corollary, decreased RsxA would lead to increased expression of the SoxR regulon, which would allow for more efficient response to oxidative stress, and increased fitness in the presence of sub-MIC TOB. We propose that when Tgt/Q levels in tRNA increase, RsxA synthesis is low and active SoxR levels are high, facilitating the bacterial response to aminoglycoside-dependent oxidative stress. Created with BioRender.com.

increased DNA repair efficiency, although the results do not exclude another mechanism underlying the UV phenotype, such as oxidative stress response.

In *E. coli*, it was previously proposed that overexpression of *tgt* could be linked with UV resistance (*Morgante et al., 2015*). We also analyzed tyrosine codon usage for the DNA repair genes in *E. coli*, and did not observe the same bias (*Figure 7C*), with 51% TAT bias, i.e., the expected level for a random group of genes of the *E. coli* genome, and with a TAC bias for *recOR* and *recA* (red arrows) and strong TAT bias for *lexA* (blue arrow) (*Figure 7—figure supplement 1B*, whole genome *E. coli*). These genes thus show the exact opposite bias in *E. coli* (*Figure 7C*). Unlike for *V. cholerae*, *E. coli* Δ*tgt* mutant did not show increased UV resistance (*Figure 7D*). This is consistent with the hypothesis that modification-tuned translation of codon-biased transcripts can be an additional means of regulation building upon already described and well-characterized transcriptional regulation pathways.

## Discussion

We show here that Q34 modification levels can be dynamic in bacteria and respond to external conditions; and that Q34 levels on *V. cholerae* tRNA^Tyr correlate with *tgt* expression. This is clearer in conditions where *tgt* transcription is highly induced (Δ*crp*), and more variable in conditions where this induction is low (tobramycin). As summarized in *Figure 8*, we propose that exposure to aminoglycosides increases *tgt* expression in *V. cholerae*, and impacts the decoding of tyrosine codons. As a consequence, transcripts with biased tyrosine codon usage are differentially translated. One such transcript codes for RsxA, an anti-SoxR factor. SoxR controls a regulon involved in oxidative stress response and sub-MIC aminoglycosides trigger oxidative stress in *V. cholerae* (*Baharoglu et al., 2013*), pointing to an involvement of oxidative stress response in the response to sub-MIC tobramycin stress. A link between Q34 and oxidative stress has previously been found in eukaryotic organisms (*Nagaraja et al., 2021*). Note that our results do not exclude the involvement of additional Q-regulated translation of other transcripts in the response to tobramycin. Q34 modification leads to reprogramming of the whole proteome, not only for other transcripts with codon usage bias, but also through an impact on the levels of stop codon readthrough and mistranslation at specific codons, as supported by our data.

In the tested conditions, we observe more efficient decoding of TAT vs TAC codons in the absence of Q, in *V. cholerae*. This is consistent with findings in human tRNAs, where the presence of Q34 increases translation of transcripts biased in C-ending codons (*Huber et al., 2022*). However, the opposite was shown in *E. coli* regarding tyrosine codon decoding efficiency (*Díaz Rullo and González Pastor, 2023*). Recent studies in eukaryotes also indicate slower translation of U-ending codons in the absence of Q34 (*Cirzi et al., 2023*; *Kulkarni et al., 2021*; *Tuorto et al., 2018*). It's important to note here, that in *V. cholerae* Δ*tgt*, better decoding of U-ending codons is observed only with tyrosine, and not with the other three NAC/U codons (histidine, aspartate, asparagine). This is interesting because it suggests that what we observe with tyrosine here may not adhere to a general rule about the decoding efficiency of U- or C-ending codons, but instead seems to be specific to tRNA^Tyr, at least in the context of *V. cholerae*. Such exceptions may also exist in other organisms. For example, in human cells, Q34 increases efficiency of decoding for U-ending codons and slows decoding of C-ending codons except for AAC, i.e., tRNA^Asn (*Zhao et al., 2023*). In mammalian cells (*Tuorto et al., 2018*), ribosome pausing at U-ending codons is strongly seen for Asp, His, and Asn, but less with Tyr. In *Trypanosoma brucei* (*Kulkarni et al., 2021*), reporters with a combination of the 4 NAC/NAU codons for Asp, Asn, Tyr, His have been tested, showing slow translation at U-ending version of the reporter in the absence of Q, but the effect on individual codons (e.g. Tyr only) was not tested. In mice (*Cirzi et al., 2023*), ribosome slowdown is seen for the Asn, Asp, His U-ending codons but not for the Tyr U-ending codon. In summary, Q34 generally increases decoding efficiency of U-ending codons in multiple organisms, but there appears to be additional parameters which affect tyrosine UAU decoding, at least in *V. cholerae*. Additional factors such as mRNA secondary structures or mistranslation may also contribute to the better translation of UAU versions of tested genes. Mistranslation could be an important factor. If codon decoding fidelity impacts decoding speed, then mistranslation could also contribute to decoding efficiency of Tyr UAU/UAC codons and proteome composition.

Overall, our findings are in accordance with the concept of the so-called MoTTs (*Endres et al., 2015*). We show that in *V. cholerae*, a proteins' codon content can influence its translation in a Q34 modification-dependent way, and that this can also impact the translation of antibiotic resistance

genes (here β-lactamase). Finally, we show that we can predict in silico, candidates for which translation can be modulated by the presence or absence of Q34 modification (e.g. DNA repair genes), which was confirmed using phenotypic tests (UV resistance).

Essential/housekeeping genes are generally TAC biased (*Figure 7—figure supplement 1A and B*), as well as ribosomal proteins, which carry mostly tyrosine TAC codons in both *V. cholerae* and *E. coli*. It has been proposed that codon bias corresponding to abundant tRNAs at such highly expressed genes guarantees their proper expression and avoids titration of tRNAs, allowing for efficient expression of the rest of the proteome (*Frumkin et al., 2018*). Induction of *tgt* by stress could also possibly be a signal for the cell to favor the synthesis of essential factors. Our results are also consistent with the fact that synonymous mutations can influence the expression of genes (*Kudla et al., 2009*).

Studies, mostly in eukaryotes, reveal that tRNA modifications are dynamic and not static as initially thought (*Chan et al., 2010*; *Chan et al., 2012*; *Torrent et al., 2018*). Modification levels depend on growth (*Keith et al., 1976*; *Moukadiri et al., 2014*), environmental changes (*Frey et al., 1988*), and stress (reviewed in *Barraud and Tisné, 2019*). Stress-regulated tRNA modification levels have an impact on the translation of regulators, which in turn trigger translational reprogramming and optimized responses to stress (*Galvanin et al., 2020*; *Persson, 1993*; *Pollo Oliveira and de Crécy Lagard, 2019*). We show here that *tgt* expression is regulated by tobramycin, the stringent response and the carbohydrate utilization regulator CRP, and that tRNA$^{Tyr}$ Q34 modification levels increase with *tgt* expression. The fact that such correlation between *tgt* expression and Q34 levels does not occur for all tRNAs (e.g. tRNA$^{Asp}$) indicates that other parameters also influence Q34 modification levels. One possibility is that other modifications, such as those on the anticodon loop of tRNA$^{Tyr}$, may influence the way Tgt modifies these tRNAs, as documented for other modification circuits (*Ehrenhofer-Murray, 2017*; *Han and Phizicky, 2018*). Tgt may also bind these tRNAs differently (for a review on modification specificity; *Barraud and Tisné, 2019*).

Our results also demonstrate that we can identify other Q-dependent MoTT candidates using in silico codon usage analysis. In fact, since we now have extensively calculated the codon usage biases at all codons for *V. cholerae* and *E. coli* genes, this approach is readily adaptable to any tRNA modification for which we know the differentially translated codons. Such regulation may be a possible way to tune the expression of essential or newly acquired genes, differing in GC content. It may also, in some cases, explain antibiotic resistance profiles in bacterial collections with established genome sequences, and for which observed phenotypic resistance does not always correlate with known resistance factors (*Oprea et al., 2020*). Further studies are needed to characterize the determinants of tRNA modification-dependent translational reprogramming.

## Ideas and speculation

Q34 modification is dynamically regulated. How tobramycin increases *tgt* expression remains an open question. Since there is a correlation between tobramycin and stringent response-dependent induction of *tgt*, one hypothesis could be that tobramycin induces *tgt* through stringent response activation. The stringent response is usually triggered upon starvation, e.g., when amino acids are scarce. *tgt* expression was recently shown to be regulated by tyrosine levels and to affect tRNA$^{Tyr}$ codon choice in *T. brucei* (*Dixit et al., 2021*). Tyrosine import into cells occurs through the TyrP transporter (*Whipp and Pittard, 1977*). Note that in *V. cholerae*, sub-MIC TOB strongly decreases *tyrP* (VCA0772) expression (*Carvalho et al., 2021a*), and thus likely decreases tyrosine intake. We could not detect the TyrP protein in our proteomics data, we can thus not compare TyrP levels in Δ*tgt* vs WT. We wondered whether supplementation with tyrosine or TyrP overexpression would reverse the Δ*tgt* TOB-sensitive fitness phenotype, but we observed no notable difference (not shown). Nevertheless, TyrP downregulation could be a signal mimicking tyrosine shortage and inducing *tgt*, leading to more efficient translation of TAC-biased proteins, such as TyrP itself.

Regarding CRP, the carbon catabolite regulator, it represses transcription of *tgt*. Interestingly, *V. cholerae crp* carries only TAC codons for tyrosine, and is strongly downregulated in Δ*tgt* in our Ribo-seq data, while its transcription levels remain unchanged. The downregulation of CRP translation when Q34 modifications are low (as in the Δ*tgt* strain) could be a way to de-repress *tgt* and increase Q34 modification levels. Note that CRP is involved in natural competence of *V. cholerae*, during growth on crustacean shells where horizontal gene transfer occurs. One can thus speculate that during exogenous DNA uptake, *tgt* repression by CRP could lead to better decoding of AT-rich

(i.e. TAT-biased) mRNAs. Thus, modulation of *tgt* levels during natural transformation may modulate the expression of horizontally transferred genes, which by definition may bear different GC content and codon usage. Moreover, if *tgt* expression is repressed by CRP during competence state, this would favor the translation of TAT-biased DNA repair genes and possibly recombination of incoming DNA into the chromosome. Translational reprogramming in response to DNA damage can thus be an advantageous property selected during evolution.

*V. cholerae* is the model organism for different species of *Vibrio*. We have previously shown that *V. cholerae's* response to sub-MIC antibiotic stress can be applied to other Gram-negative pathogens (*Baharoglu et al., 2013*; *Gutierrez et al., 2013*), while there are differences between *E. coli* and *V. cholerae*, in response to sub-MIC antibiotics and oxidative stress phenotypes (*Baharoglu et al., 2013*; *Baharoglu and Mazel, 2011*; *Gutierrez et al., 2013*). Here, we have also addressed some of the effects of TOB in *E. coli Δtgt* mutant. In *E. coli,* the deletion of *tgt* has a less dramatic effect on the susceptibility to TOB (*Babosan et al., 2022* and *Figure 5—figure supplement 2*). *V. cholerae* and *E. coli* globally show similar tyrosine codon usage in their genomes (*Figure 7—figure supplement 1A and B*). However, *E. coli rsxA* does not display a codon bias toward TAT, and neither do DNA repair genes. One can think that in regard to MoTTs, different organisms have evolved according to the environments where they grow, selecting the integration of specific stress response pathways under specific post-transcriptional regulations.

It was recently shown that *E. coli Δtgt* strain is more resistant than the WT to nickel toxicity, most certainly because the nickel importer genes *nikABCDE* are less expressed, but the underlying molecular mechanism had not been elucidated (*Pollo-Oliveira et al., 2022*). As NikR, the repressor of the *nik* operon is enriched in TAT codons (100%), a more efficient translation of the *nikR* gene in the absence of Q34 would lead to the observed repression phenotype. In addition, the nickel exporter gene *rcnA* is also enriched is TAT (100%), while one of the genes for subunits of the nickel importer *nikD* is enriched in TAC codons (100%). In combination this could explain the clear resistance of the *tgt* strain to high levels of nickel.

However, protein levels are not always in line with the codon bias predictions. The positions of the codons of interest and their sequence context may also be important for differential translation. There could be an interplay or synergies between different codons decoded by Q-modified tRNAs. The presence of the codons of interest in the 5'-end vs 3'-end of a transcript could have a bigger impact on the efficiency of translation (*Boël et al., 2016*; *Osterman et al., 2020*). A recent study testing TAC/TAT codons placed between two genes in a translational fusion yielded different results compared to our constructs with the tested codons at the 5' of the transcript (*Kimura et al., 2022*). Similarly, the distance between two codons of interest or the identity of the nearby codons may be important. The translation of highly transcribed genes and genes with low levels of mRNAs could be dissimilar. Codon usage may also directly impact gene expression at mRNA levels with an effect on transcription termination (*Zhao et al., 2021*), especially for constitutive genes. Thus, the search for MoTTs could be facilitated by comparing transcriptomics to proteomics data, and additional experiments need to be performed to elucidate post-transcriptional regulation-related phenotypes, but the differential expression of specific TAT/TAC-biased proteins finally allows to propose a model for the pleiotropic phenotype caused by Q34 deficiency in *E. coli*.

Finally, the presence or absence of a modification could also affect aminoacylation of the tRNA. Both TAT and TAC codons are decoded by the same and only one tRNA, tRNA$^{TYR}_{GUA}$. In this case, a defect in aminoacylation of this tRNA would impact the decoding of both codons. Our results do not support such an aminoacylation problem, because the efficiency of decoding decreases for one codon (TAC) and not for the other (TAT) in *Δtgt*, and the difference is clearer in TOB. These results are also consistent with rescue of TOB-sensitive phenotype by tRNA$^{Tyr}_{GUA}$ but not tRNA$^{Tyr}_{AUA}$ overexpression in *Δtgt*.

This study, and others mentioned in the introduction, indicate that stress-regulated tRNA modifications can facilitate homeostasis by reprogramming the translation of stress response genes. The diversity of tRNA modifications, their specific effects on various proteins, and stress responses thus make them a promising field of study.

## Materials and methods

### Media and growth conditions

Platings were done at 37°C, in Mueller-Hinton (MH) agar media. Liquid cultures were grown at 37°C in MH in aerobic conditions, with 180 rotations per minute.

Competition experiments were performed as described (*Babosan et al., 2022*): overnight cultures from single colonies of mutant *lacZ+* and WT *lacZ-* strains were washed in PBS (phosphate-buffered saline) and mixed 1:1 (500 µl+500 µL). At this point 100 µL of the mix were serial diluted and plated on MH agar supplemented with X-gal (5-bromo-4-chloro-3-indolyl-β-D-galactopyranoside) at 40 µg/mL to assess T0 initial 1:1 ratio. At the same time, 10 µL from the mix were added to 2 mL of MH or MH supplemented with sub-MIC antibiotics (concentrations, unless indicated otherwise: TOB: tobramycin 0.6 µg/mL; GEN: 0.5 µg/mL; CIP: ciprofloxacin 0.01 µg/mL, CRB: carbenicillin 2.5 µg/mL), PQ: paraquat 10 µM, or $H_2O_2$: 0.5 mM. Cultures were incubated with agitation at 37°C for 20 hr, and then diluted and plated on MH agar plates supplemented with X-gal. Plates were incubated overnight at 37°C and the number of blue and white CFUs was assessed. Competitive index was calculated by dividing the number of blue CFUs (*lacZ+* strain) by the number of white CFUs (*lacZ-* strain) and normalizing this ratio to the T0 initial ratio. When a plasmid was present, antibiotic was added to maintain selection: kanamycin 50 µg/mL for pSEVA.

### Construction of complementation and overexpression plasmids in pSEVA238

Genes were amplified on *V. cholerae* genomic DNA using primers listed in *Supplementary file 4* and cloned into pSEVA238 (*Silva-Rocha et al., 2013*) under the dependence of the *Pm* promoter (*Kessler et al., 1994*), by restriction digestion with XbaI+EcoRI and ligation using T4 DNA ligase. Promoter *Pm* is originally derived from the *Pseudomonas putida* toluate catabolic operon and is positively regulated by the benzoate-inducible XylS transcriptional regulator. Sodium benzoate 1 mM was added in the medium as inducer.

Survival/tolerance tests were performed on early exponential phase cultures. The overnight stationary phase cultures were diluted 1000× and grown until OD 600 nm=0.35–0.4, at 37°C with shaking, in Erlenmeyers containing 25 mL fresh MH medium. Appropriate dilutions were plated on MH plates to determine the total number of CFUs in time zero untreated cultures. 5 mL of cultures were collected into 50 mL Falcon tubes and treated with lethal doses of desired antibiotics (5 or 10 times the MIC: tobramycin 5 or 10 µg/mL, carbenicillin 50 µg/mL, ciprofloxacin 0.025 µg/mL) for 30 min, 1 hr, 2 hr, and 4 hr if needed, at 37°C with shaking in order to guarantee oxygenation. Appropriate dilutions were then plated on MH agar without antibiotics and proportion of growing CFUs were calculated by doing a ratio with total CFUs at time zero. Experiments were performed three to eight times.

### MIC determination

Stationary phase cultures grown in MH were diluted 20 times in PBS, and 300 µL were plated on MH plates and dried for 10 min. ETEST straps (Biomérieux) were placed on the plates and incubated overnight at 37°C.

Quantification of fluorescent neomycin uptake was performed as described (*Lang et al., 2021*). Neo-Cy5 is the neomycin aminoglycoside coupled to the fluorophore Cy5, and has been shown to be active against Gram-bacteria (*Okuda, 2015*; *Sabeti Azad et al., 2020*). Briefly, overnight cultures were diluted 100-fold in rich MOPS (Teknova EZ-rich defined medium). When the bacterial strains reached an OD 600 nm of ~0.25, they were incubated with 0.4 µM of Cy5 labeled neomycin for 15 min at 37°C. 10 µL of the incubated culture were then used for flow cytometry, diluting them in 250 µL of PBS before reading fluorescence. WT *V. cholerae* was incubated simultaneously without neo-Cy5 as a negative control. Flow cytometry experiments were performed as described (*Baharoglu et al., 2010*) and repeated at least three times. For each experiment, 100,000 events were counted on the Miltenyi MACSquant device.

### PMF measurements

Quantification of PMF was performed using the MitoTracker Red CMXRos dye (Invitrogen) as described (*El Mortaji et al., 2020*), in parallel with the neo-Cy5 uptake assay, using the same bacterial cultures.

50 µL of each culture were mixed with 60 µL of PBS. Tetrachlorosalicylanilide (TCS) (Thermo Fisher), a protonophore, was used as a negative control with a 500 µM treatment applied for 10 min at room temperature. Then, 25 nM of MitoTracker Red were added to each sample and let at room temperature for 15 min under aluminum foil. 20 µL of the treated culture were then used for flow cytometry, diluted in 200 µL of PBS before reading fluorescence.

## tRNA overexpressions

Synthetic fragments carrying the P*trc* promoter, the desired tRNA sequence, and the natural transcriptional terminator sequence of VCt002 were ordered from IDT as double-stranded DNA gBlocks, and cloned into pTOPO plasmid. Sequences are indicated in *Supplementary file 4*.

## mRNA purification

For RNA extraction, overnight cultures were diluted 1:1000 in MH medium and grown with agitation at 37°C until an OD 600 of 0.3–0.4 (exponential phase). 0.5 mL of these cultures were centrifuged and supernatant removed. Pellets were homogenized by resuspension with 1.5 mL of room temperature TRIzol Reagent. Next, 300 µL chloroform were added to the samples following mix by vortexing. Samples were then centrifuged at 4°C for 10 min. Upper (aqueous) phase was transferred to a new 2 mL tube and mixed with 1 volume of 70% ethanol. From this point, the homogenate was loaded into a RNeasy Mini kit (QIAGEN) column and RNA purification proceeded according to the manufacturer's instructions. Samples were then subjected to DNase treatment using TURBO DNA-free Kit (Ambion) according to the manufacturer's instructions.

## mRNA quantifications by digital-RT-PCR

qRT-PCRs were prepared with 1 µL of diluted RNA samples using the qScript XLT 1-Step RT-qPCR ToughMix (Quanta Biosciences, Gaithersburg, MD, USA) within Sapphire chips. Digital PCR was conducted on a Naica Geode programmed to perform the sample partitioning step into droplets, followed by the thermal cycling program suggested in the user's manual. Primer and probe sequences used in digital qRT-PCR are listed in *Supplementary file 4*. Image acquisition was performed using the Naica Prism3 reader. Images were then analyzed using Crystal Reader software (total droplet enumeration and droplet quality control) and the Crystal Miner software (extracted fluorescence values for each droplet). Values were normalized against expression of the housekeeping gene *gyrA* as previously described (*Lo Scrudato and Blokesch, 2012*).

## tRNA level quantification by qRT-PCR

First-strand cDNA synthesis and quantitative real-time PCR were performed with KAPA SYBR FAST Universal (CliniSciences) on the QuantStudio Real-Time PCR (Thermo Fisher) using the primers indicated in *Supplementary file 4*. Transcript levels of each gene were normalized to *gyrA* as the reference gene control (*Lo Scrudato and Blokesch, 2012*). Gene expression levels were determined using the $2^{-\Delta\Delta Cq}$ method (*Bustin et al., 2009*; *Livak and Schmittgen, 2001*) in respect to the MIQE guidelines. Relative fold-difference was expressed either by reference to antibiotic free culture or the WT strain in the same conditions. All experiments were performed as three independent replicates with all samples tested in duplicate. Cq values of technical replicates were averaged for each biological replicate to obtain the ΔCq. After exponential transformation of the ΔCq for the studied and the normalized condition, medians and upper/lower values were determined.

## Stop codon readthrough quantification assay

*V. cholerae* electrocompetent cells were transformed with dual reporter plasmids that were previously described (*Fabret and Namy, 2021*). Overnight cultures of transformants were diluted 1:100 in MH medium supplemented with 5 µg/mL chloramphenicol to maintain plasmids, 200 µg/mL IPTG (isopropyl β-D-1-thiogalactopyranoside) and in the presence or not of tobramycine 0.4 µg/mL and grown with shaking at 37°C until an OD 600 of 0.3 (exponential phase).

Luciferase luminescence was quantified using the luciferase assay system (Promega, WI, USA, Cat.# E1500). Briefly, 90 µL of each culture were aliquoted in 1.5 mL tubes, treated with 10 µL $K_2HPO_4$ (1 M) and EDTA (20 mM) and quick-frozen in dry ice for 1 min. Tubes were then placed in room temperature water for 5 min to allow the cultures to thaw. 300 µL of lysis buffer (Cell Culture Lysis Reagent 1X;

lysozyme 1.25 mg/mL; BSA 2.5 mg/mL) were added in the tubes that were then placed back in water for 10 min. 100 µL of lysate were placed in 5 mL tubes with 100 µL of Luciferase Assay Reagent and luminescence was read for 10 s using Lumat LB 9507 (EG&G Berthold).

For β-galactosidase activity quantification, 2 mL of the cultures were aliquoted and mixed with 50 µL chloroform and 50 µL SDS 0.1%. After vortexing for 45 s, samples were placed 5 min at room temperature for cell lysis. 500 µL of the lysates were collected into 5 mL tubes and treated with 1.5 mL Z-Buffer (8.5 mg/mL $Na_2HPO_4$; 5.5 mg/mL $NaH_2PO_4H_2O$; 0.75 mg/mL KCl; 0.25 mg/mL $MgSO_4$, $7H_2O$) supplemented with 7 µL/mL 2-mercaptoethanol. After 5 min incubation at 37°C, 500 µL ONPG (4 mg/mL) were added in the samples which were then placed at 37°C for 1 hr. Reaction has finally been stopped by addition on the tubes of 1 mL $Na_2CO_3$ (1 M). 2 mL suspension were transferred to Eppendorf tubes, centrifuged and OD 420 nm of the supernatant was read. β-Galactosidase activity dosage was used for normalization of luminescence.

## Construction of gfp reporters where tyrosine 66 chromophore was replaced with another codon

Whole plasmid amplifications were performed on pSC101-gfp using primers introducing the desired point mutation at codon position 66 of GFP. An example is given below for the primers replacing tyr TAT with his CAT.

Forward primer: CACTACTTTCGGT**CAT**GGTGTTCAATGCTTTGCG. Reverse primer: TGAACACC**ATG**ACCGAAAGTAGTGACAAGTGTTGG. PCR: 30 cycles, annealing temperature 55°C, elongation time, 10 min.

## Construction of *gfp* reporters with codon stretches

The positive control was *gfp*mut3 (stable *gfp*) (**Cormack et al., 1996**) under the control of P*trc* promoter, the transcription start site, *rbs,* and ATG start codon are indicated in bold and underlined.

TTGACAATTAATCATCCGGCTCGTATAATGTGTGG<u>A</u>ATTGTGAGCGGATAACAATTTCACAC
<u>AGGAAAC</u>AGCGCCGC<u>ATG</u>CGTAAAGGAGAAGAACTTTTCACTGGAGTTGTCCCAATTCTTGT
TGAATTAGATGGTGATGTTAATGGGCACAAATTTTCTGTCAGTGGAGAGGGTGAAGGTGATGCA
ACATACGGAAAACTTACCCTTAAATTTATTTGCACTACTGGAAAACTACCTGTTCCATGGCCAA
CACTTGTCACTACTTTCGGTTATGGTGTTCAATGCTTTGCGAGATACCCAGATCATATGAAACAGCAT
GACTTTTTCAAGAGTGCCATGCCCGAAGGTTATGTACAGGAAAGAACTATATTTTTCAAAGATG
ACGGGAACTACAAGACACGTGCTGAAGTCAAGTTTGAAGGTGATACCCTTGTTAATAGAATCGA
GTTAAAAGGTATTGATTTTAAAGAAGATGGAAACATTCTTGGACACAAATTGGAATACAACTATAACT
CACACAATGTATACATCATGGCAGACAAACAAAAGAATGGAATCAAAGTTAACTTCAAAATTAG
ACACAACATTGAAGATGGAAGCGTTCAACTAGCAGACCATTATCAACAAAATACTCCAATTGGC
GATGGCCCTGTCCTTTTACCAGACAACCATTACCTGTCCACACAATCTGCCCTTTCGAAAGATC
CCAACGAAAAGAGAGACCACATGGTCCTTCTTGAGTTTGTAACAGCTGCTGGGATTACACATGG
CATGGATGAACTATACAAATAA.

For the tested codon stretches, six repeats of the desired codon were added just after the ATG start codon of *gfp*. The DNA fragments were ordered as double-stranded *eblocks* from Integrated DNA Technologies (IDT), and cloned into pTOPO-Blunt using kanamycin resistance, following the manufacturer's instructions.

For tests of sequence context surrounding tyrosine codons of *rsxA*, DNA was ordered from IDT and cloned into pTOPO as described for codon stretches above, based on the following amino acid sequences (tested sequences in bold):

## VC1017RsxA *V. cholerae*

MLLLWQSRIMPGSEANIYITM<u>TEYLLL</u>LIGTVLVNNFVLVKFLGLCPFMGVSKKLETAIGMGLATTFVLTL
ASVCAYLVESYVLRPLGI<u>EYLR</u>TMSFILVIAVVVQFTEMVVHKTSPT<u>LYRLL</u>GIFLPLITTNCAVLGVALLNINEN
HNFIQSIIYGFGAAVGFSLVLILFASMRERIHVADVPAPFKGASIAMITAGLMSLAFMGFTGLVKL.

## RsxA *E. coli*

M<u>TDYLLL</u>FVGTVLVNNFVLVKFLGLCPFMGVSKKLETAMGMGLATTFVMTLASICAWLIDTWILIPLNLIY
LRTLAFILVIAVVVQFTEMVVRKTSPV<u>LYRLL</u>GIFLPLITTNCAVLGVALLNINLGHNFLQSALYGFSAAVGFSLV
MVLFAAIRERLAVADVPAPFRGNAIALITAGLMSLAFMGFSGLVKL.

## Quantification of *gfp* fusion expression by fluorescent flow cytometry

Flow cytometry experiments were performed as described (*Baharoglu et al., 2010*) on overnight cultures and repeated at least three times. For each experiment, 50,000–100,000 events were counted on the Miltenyi MACSquant device. The mean fluorescence per cell was measured at the FITC channel for each reporter in both WT and Δ*tgt* strains, and the relative fluorescence was calculated as the ratio of the mean fluorescence of a given reporter in Δ*tgt* over the mean fluorescence of the same reporter in the WT. Native *gfp* (*gfpmut3*) was used as control.

Transcriptional fusion: *rsxA* promoter sequence was amplified using primers ZIP796/ZIP812. gfp was amplified from pZE1-*gfp* (*Elowitz and Leibler, 2000*) using primers ZIP813/ZIP200. The two fragments were PCR assembled into *PrsxA-gfp* using ZIP796/ZIP200 and cloned into pTOPO-TA cloning vector. The *PrsxA-gfp* fragment was then extracted using *EcoRI* and cloned into the low copy plasmid pSC101 (1–5 copies per cell). The plasmid was introduced into desired strains, and fluorescence was measured on indicated conditions, by counting 100,000 cells on the Miltenyi MACSquant device. Likewise, the control plasmid *Pc-gfp* (constitutive) was constructed using primers ZIP513/ZIP200 and similarly cloned in pSC101.

For translational fusions, the constitutive P*trc* promoter, the *rsxA* gene (without stop codon) with desired codon usage fused to *gfp* (without ATG start codon) was ordered from IDT in the pUC-IDT vector (carbenicillin resistant).

Native sequence of *V. cholerae rsxA* gene, called *rsxA*^TAT^*gfp* in this manuscript, is shown below. For *rsxA*^TAC^*gfp,* all tyrosine TAT codons were replaced with TAC.

ATGACCGAA**TAT**CTTTTGTTGTTAATCGGCACCGTGCTGGTCAATAACTTTGTACTGGTGAAGT
TTTTGGGCTTATGTCCTTTTATGGGCGTATCAAAAAAACTAGAGACCGCCATTGGCATGGGGGTT
GGCGACGACATTCGTCCTCACCTTAGCTTCGGTGTGCGCT**TAT**CTGGTGGAAAGT**TAC**GTGTTACG
TCCGCTCGGCATTGAG**TAT**CTGCGCACCATGAGCTTTATTTTGGTGATCGCTGTCGTAGTACAGTTC
ACCGAAATGGTGGTGCACAAAACCAGTCCGACACTC**TAT**CGCCTGCTGGGCATTTTCCTGCCA
CTCATCACCACCAACTGTGCGGTATTAGGGGTTGCGCTGCTCAACATCAACGAAAATCACAACT
TTATTCAATCGATCATT**TAT**GGTTTTGGCGCTGCTGTTGGCTTCTCGCTGGTGCTCATCTTGTTCGCT
TCAATGCGTGAGCGAATCCATGTAGCCGATGTCCCCGCTCCCTTTAAGGGCGCATCCATTGCGA
TGATCACCGCAGGTTTAATGTCTTTGGCCTTTATGGGCTTTACCGGATTGGTGAAACTGGCTAGC.

*gfp*^TAC^ and *gfp*^TAT^ (tyrosine 11 TAT instead of 11 TAC) were ordered from IDT as synthetic genes under the control of P*trc* promoter in the pUC-IDT plasmid (carbenicillin resistant). The complete sequence of ordered fragments is indicated in *Supplementary file 4*, tyrosine codons are underlined.

## Construction of *bla* reporters

Point mutations for codon replacements were performed using primer pairs where the desired mutations were introduced and by whole plasmid PCR amplification on circular pTOPO-TA plasmid. Primers are listed in *Supplementary file 4*.

## Tolerance tests with *bla* reporters

A single colony from fresh transformation plates was inoculated in 24-well plates, each well containing 2 mL of MH. Cells were grown to early exponential phase without carbenicillin, and with or without tobramycin 20% of MIC (TOB 0.2 µg/mL). After 2 hr of incubation at 37°C with shaking (early exponential phase), dilutions were spotted in parallel on plates with or without carbenicillin (time T0). Cultures were then treated with carbenicillin at 10x MIC (50 µg/mL) for 20 hr, at 37°C with shaking. Dilutions were spotted on plates with or without carbenicillin. Surviving cells shown here are sensitive to carbenicillin (no growth on carbenicillin containing plates), suggesting that increased or decreased survival was due to increased (erroneous translation) or decreased (faithful translation) β-lactamase activity at the time of treatment.

## Growth on microtiter plate reader for *bla* reporter assays

Overnight cultures were diluted 1:500 in fresh MH medium, on 96-well plates. Each well contained 200 µL. Plates were incubated with shaking on TECAN plate reader device at 37°C, OD 600 nm was measured every 15 min. Tobramycin was used at sub-MIC: TOB 0.2 µg/mL. Kanamycin and carbenicillin were used at selective concentration: CRB 100 µg/mL, KAN 50 µg/mL.

## Protein extraction

Overnight cultures of *V. cholerae* were diluted 1:100 in MH medium and grown with agitation at 37°C until an OD 600 nm of 0.3 (exponential phase). 50 mL of these cultures were centrifuged for 10 min at 4°C and supernatant removed. Lysis was achieved by incubating cells in the presence of lysis buffer (10 mM Tris-HCl pH 8, 150 mM NaCl, 1% Triton X-100) supplemented with 0.1 mg/mL lysozyme and complete EDTA-free Protease Inhibitor Cocktail (Roche) for 1 hr on ice. Resuspensions were sonicated 3×50 s (power: 6, pulser: 90%), centrifuged for 1 hr at 4°C at 5000 rpm and supernatants were quantified using Pierce BCA Protein Assay Kit (Cat. No 23225) following the manufacturer's instructions. Proteins were then stored at –80°C.

## Proteomics MS and analysis

### Sample preparation for MS

Tryptic digestion was performed using eFASP (enhanced Filter-Aided Sample Preparation) protocol (*Erde et al., 2014*). All steps were done in 30 kDa Amicon Ultra 0.5 mL filters (Millipore). Briefly, the sample was diluted with a 8 M urea, 100 mM ammonium bicarbonate buffer to obtain a final urea concentration of 6 M. Samples were reduced for 30 min at room temperature with 5 mM TCEP. Subsequently, proteins were alkylated in 5 mM iodoacetamide for 1 hr in the darkness at room temperature and digested overnight at 37°C with 1 µg trypsin (Trypsin Gold Mass Spectrometry Grade, Promega). Peptides were recovered by centrifugation, concentrated to dryness, and resuspended in 2% acetonitrile (ACN)/0.1% FA just prior to LC-MS injection.

### LC-MS/MS analysis

Samples were analyzed on a high-resolution mass spectrometer, Q Exactive Plus Hybrid Quadrupole-Orbitrap Mass Spectrometer (Thermo Scientific), coupled with an EASY 1200 nLC system (Thermo Fisher Scientific, Bremen). One µg of peptides was injected onto a home-made 50 cm C18 column (1.9 µm particles, 100 Å pore size, ReproSil-Pur Basic C18, Dr. Maisch GmbH, Ammerbuch-Entringen, Germany). Column equilibration and peptide loading were done at 900 bars in buffer A (0.1% FA). Peptides were separated with a multi-step gradient from 3% to 22% buffer B (80% ACN, 0.1% FA) in 160 min, 22% to 50% buffer B in 70 min, 50% to 90% buffer B in 5 min at a flow rate of 250 nL/min. Column temperature was set to 60°C. The Q Exactive Plus Hybrid Quadrupole-Orbitrap Mass Spectrometer (Thermo Scientific) was operated in data-dependent mode using a Full MS/ddMS2 Top 10 experiment. MS scans were acquired at a resolution of 70,000 and MS/MS scans (fixed first mass 100 m/z) at a resolution of 17,500. The AGC target and maximum injection time for the survey scans and the MS/MS scans were set to 3E6, 20 ms and 1E6, 60 ms, respectively. An automatic selection of the 10 most intense precursor ions was activated (Top 10) with a 35 s dynamic exclusion. The isolation window was set to 1.6 m/z and normalized collision energy fixed to 27 for HCD fragmentation. We used an underfill ratio of 1.0% corresponding to an intensity threshold of 1.7E5. Unassigned precursor ion charge states as well as 1, 7, 8, and >8 charged states were rejected and peptide match was disable.

### Data analysis

Acquired raw data were analyzed using MaxQuant 1.5.3.8 version (*Cox et al., 2011*) using the Andromeda search engine (*Tyanova et al., 2016*) against *V. cholerae* Uniprot reference proteome database (3782 entries, download date February 21, 2020) concatenated with usual known mass spectrometry contaminants and reversed sequences of all entries. All searches were performed with oxidation of methionine and protein N-terminal acetylation as variable modifications and cysteine carbamidomethylation as fixed modification. Trypsin was selected as protease allowing for up to two missed cleavages. The minimum peptide length was set to five amino acids. The false discovery rate (FDR) for peptide and protein identification was set to 0.01. The main search peptide tolerance was set to 4.5 ppm and to 20 ppm for the MS/MS match tolerance. One unique peptide to the protein group was required for the protein identification. An FDR cut-off of 1% was applied at the peptide and protein levels.

The statistical analysis of the proteomics data was performed as follows: three biological replicates were acquired per condition. To highlight significantly differentially abundant proteins

between two conditions, differential analyses were conducted through the following data analysis pipeline: (1) deleting the reverse and potential contaminant proteins; (2) keeping only proteins with at least two quantified values in one of the three compared conditions to limit misidentifications and ensure a minimum of replicability; (3) log2-transformation of the remaining intensities of proteins; (4) normalizing the intensities by median centering within conditions, thanks to the normalizeD function of the R package DAPAR (*Wieczorek et al., 2017*), (5) putting aside proteins without any value in one of both compared conditions: as they are quantitatively present in a condition and absent in another, they are considered as differentially abundant proteins and (6) performing statistical differential analysis on them by requiring a minimum fold-change of 2.5 between conditions and by using a LIMMA t-test (*Ritchie et al., 2015*) combined with an adaptive Benjamini-Hochberg correction of the p-values, thanks to the adjust.p function of the R package cp4p (*Giai Gianetto et al., 2016*). The robust method of Pounds and Cheng was used to estimate the proportion of true null hypotheses among the set of statistical tests (*Pounds and Cheng, 2006*). The proteins associated with an adjusted p-value inferior to an FDR level of 1% have been considered as significantly differentially abundant proteins. Finally, the proteins of interest are therefore the proteins that emerge from this statistical analysis supplemented by those being quantitatively absent from one condition and present in another. The mass spectrometry proteomics data have been deposited to the ProteomeXchange Consortium via the PRIDE partner repository with the dataset identifier PXD035297.

## RNA purification for RNA-seq

Cultures were diluted 1000× and grown in triplicate in MH supplemented or not with 0.6 µg/mL of tobramycin, corresponding to 50% of the MIC in liquid cultures, to an OD 600 nm of 0.4. RNA was purified with the RNeasy Mini kit (QIAGEN) according to the manufacturer's instructions. Briefly, 4 mL of RNA protect (QIAGEN) reagent were added on 2 mL of bacterial cultures during 5 min. After centrifugation, the pellets were conserved at –80°C until extraction. Protocol 2 of the RNA protect Bacteria Reagent Handbook was performed, with the addition of a proteinase K digestion step, such as described in the protocol 4. Quality of RNA was controlled using the Bioanalyzer. Sample collection, total RNA extraction, library preparation, sequencing, and analysis were performed as previously described (*Krin et al., 2018*). The data for this RNA-seq study has been submitted in the GenBank repository under the project number GSE214520.

## GO enrichment analysis

GO enrichment analyses were performed on http://geneontology.org/ as follows: Binomial test was used to determine whether a group of genes in the tested list was more or less enriched than expected in a reference group. The annotation dataset used for the analysis was GO biological process complete.

The analyzed lists were for each condition (MH/TOB), genes (*Supplementary file 3*) with at least twofold change in RNA-seq data of WT strain compared to *Δtgt*, and with an adjusted p-value<0.05. The total number of uploaded gene list to be analyzed were 53 genes for MH and 60 genes for TOB. The reference gene list was *V. cholerae* (all genes in database), 3782 genes. Annotation Version: PANTHER Overrepresentation Test (Released 2022-07-12). GO Ontology database DOI: 10.5281/zenodo.6399963 (Released 2022-03-22).

## Stringent response measurement

*P1rrnB-gfp* fusion was constructed using *gfp* ASV (*Andersen et al., 1998*), and cloned into plasmid pSC101. *P1rrnB-GFPasv* transcriptional fusion was amplified from strain R438 (*E. coli* MG1655 *attB::P1rrnB gfp-ASV::kan* provided by Ivan Matic) using primers AFC060 and AFC055, thus including 42 bp upstream of *rrnB* transcription initiation site. PCR product was then cloned into pTOPOblunt vector and subcloned into pSC101 by *EcoRI* digestion and ligation. The final construct was confirmed by Sanger sequencing. The plasmid was then introduced by electroporation into the tested strains. Overnight cultures were performed in MH+carbenicillin 100 µg/mL and diluted 500× in 10 mL fresh MH or MH+TOB 0.4 µg/mL, in an Erlenmeyer. At time points 0 min, and every 30 during 3 hr, the OD 600 nm was measured and fluorescence was quantified in flow cytometry. For each experiment, 50,000–100,000 events were counted on the Miltenyi MACSquant device.

## tRNA-enriched RNA extraction

Overnight cultures of *V. cholerae* were diluted 1:1000 in MH medium and grown in aerobic conditions, with 180 rotations per minute at 37°C until an OD 600 nm of 0.5. tRNA-enriched RNA extracts were prepared using room temperature TRIzol reagent as described (*Galvanin et al., 2019*) and contaminating DNA were eliminated using TURBO DNA-free Kit (Ambion). RNA concentration was controlled by UV absorbance using NanoDrop 2000c (Thermo Fisher Scientific). The profile of isolated tRNA fractions was assessed by capillary electrophoresis using an RNA 6000 Pico chip on Bioanalyzer 2100 (Agilent Technologies).

## tRNA-enriched sample digestion for quantitative analysis of queuosine by mass spectrometry

Purified tRNA-enriched RNA fractions were digested to single nucleosides using the New England BioLabs Nucleoside digestion mix (Cat. No. M0649S). 10 µL of the RNA samples diluted in ultrapure water to 100 ng/µL were mixed with 1 µL of enzyme, 2 µL of Nucleoside Digestion Mix Reaction Buffer (10×) in a final volume of 20 µL in nuclease-free 1.5 mL tubes. Tubes were wrapped with parafilm to prevent evaporation and incubated at 37°C overnight.

## Queuosine quantification by LC-MS/MS

Analysis of global levels of queuosine (Q) was performed on a Q exactive mass spectrometer (Thermo Fisher Scientific). It was equipped with an electrospray ionization source (H-ESI II Probe) coupled with an Ultimate 3000 RS HPLC (Thermo Fisher Scientific). The Q standard was purchased from Epitoire (Singapore).

Digested RNA was injected onto a Thermo Fisher Hypersil Gold aQ chromatography column (100 mm * 2.1 mm, 1.9 µm particle size) heated at 30°C. The flow rate was set at 0.3 mL/min and run with an isocratic eluent of 1% ACN in water with 0.1% formic acid for 10 min.

Parent ions were fragmented in positive ion mode with 10% normalized collision energy in parallel-reaction monitoring (PRM) mode. MS2 resolution was 17,500 with an AGC target of 2e5, a maximum injection time of 50 ms, and an isolation window of 1.0 m/z.

The inclusion list contained the following masses: G (284.1) and Q (410.2). Extracted ion chromatograms of base fragments (±5 ppm) were used for detection and quantification (152.0565 Da for G; 295.1028 Da for Q). The secondary base fragment 163.0608 was also used to confirm Q detection but not for quantification.

Calibration curves were previously generated using synthetic standards in the ranges of 0.2–40 pmol injected for G and 0.01–1 pmol for Q. Results are expressed as a percentage of total G.

## Queuosine detection by sequencing

The detection of queuosine was performed as described in *Katanski et al., 2022*. Briefly, 200 ng of total RNA were subjected to oxidation by 45 mM of $NaIO_4$ in 50 mM AcONa pH 5.2 buffer for 1 hr at 37°C. The reaction was quenched by the addition of 36 mM glucose and incubation for 30 min at 37°C and RNA was precipitated with absolute ethanol. After precipitation and two washes by 80% ethanol, the RNA pellet was resuspended in 3' ligation reaction buffer 1× and subjected to library preparation using the NEBNext Small RNA Library Prep Set for Illumina (NEB, #E7330S). Specific primers for *V. cholerae* tRNAAsn_GUU1, tRNAAsn_GUU2, tRNAAsp_GUC, tRNATyr_GUA, and tRNAHis_GUG were hybridized instead of RT primer used in NEBNext Small RNA Library Prep kit, under the same hybridization conditions. The 5'-SR adaptor was ligated, and reverse transcription was performed for 1 hr at 50°C followed by 10 min at 80°C using Superscript IV RT (instead of Protoscript II used in the kit). PCR amplification was performed as described in the manufacturer's protocol. Libraries were qualified using Tapestation 4150 and quantified using Qubit fluorometer. Libraries were multiplexed and sequenced in a 50 bp single read mode using NextSeq2000 (Illumina, San Diego, CA, USA).

Bioinformatic analysis was performed by trimming of raw reads using trimmomatic v0.39 to remove adapter sequences as well as very short and low-quality sequencing reads. Alignment was done by bowtie2 (v2.4.4) in End-to-End mode with `--mp` *2* `--rdg` *0,2* options to favor retention of reads with deletions, only non-ambiguously mapped reads were taken for further analysis. Coverage file was created with *samtools mpileup* and deletion signature extracted for every position using custom R script. Deletion score was calculated as number of deletions divided by number of matching nucleotides at

a given position. Analysis of Q tRNA modification in *V. cholerae* strains was performed in triplicate for biological replicates with technical duplicate for each sample.

## Analysis of queuosine tRNA modification using APB northern blot assay

Quantification of queuosine in tRNA-Tyr from purified tRNA-enriched RNA fractions was performed using a non-radioactive northern blot method: the procedure for pouring and running APB gels was based on the method detailed in *Cirzi and Tuorto, 2021*. tRNA-Tyr were detected using the following 3'-end digoxigenin (DIG)-labeled probe: 5' - CTTTGGCCACTCGGGAACCCCTCC - 3'DIG.

For 1 gel, ABP gel buffer was prepared by mixing 4.2 g urea, 50 mg 3-(Acrylamido) phenylboronic acid (Sigma-Aldrich, Cat. No. 771465), 1 mL 10× RNase-free TAE, 3.2 mL 30% acrylamide and bis-acrylamide solution 37.5:1 and adding water to adjust the final volume to 10 mL. After stirring to facilitate dissolution and right before pouring, 10 µL TEMED and 60 µL 10% APS were added to the 10 mL ABP buffer to catalyze and initiate polymerization respectively. Gels were casted using the Mini-PROTEAN Bio-Rad handcast system, short plates (70×100 mm$^2$), 0.75 mm spacers and 10-well gel combs. Gels were left to polymerize at room temperature for 50 min.

Alkaline hydrolysis in 100 mM of Tris-HCl pH 9 of our tRNA-enriched RNA extracts was carried out at 37°C for 30 min to break the ester bonds between tRNAs and their cognate amino acids. 10 µL of the deacylated tRNA-enriched RNA samples were mixed with 8 µL of 2× RNA loading dye (Thermo Scientific Cat. No. R0641), denatured for 3 min at 72°C and the whole volume was loaded onto the gel. Electrophoresis of the gels were carried out in the Mini-PROTEAN Tetra Vertical Electrophoresis Cell for 30 min at 85 V at room temperature followed by 1h30 at 140 V at 4°C. Gels were incubated at room temperature for 15 min with shaking in 50 mL 1× RNAse-free TAE mixed with 10 µL of 10,000× SYBR Gold nucleic acid staining solution (Invitrogen S11494) and nucleic acids were visualized using a transilluminator. Transfers of the nucleic acids to positively charged nylon membranes were performed at 5 V/gel for 40 min at room temperature using a semi-dry blotting system. RNAs were crosslinked to the membrane surface through exposure to 254 nm UV light at a dose of 1.2 J.

Membranes were transferred into glass bottles containing 5 mL of pre-warmed hybridization buffer and incubated for 1 hr at 42°C at a constant rotation in a hybridization oven. Hybridization buffer was obtained by mixing 12.3 mL of 20× SSX, 1 mL of 1 M Na$_2$HPO$_4$ pH 7.2, 17.5 mL of 20% SDS, 2 mL of 50× Denhardt's solution, and 17 mL RNase-free H$_2$O. 3 µL of the DIG-labeled probe solution at 100 pmol/µL were then added into the 5 mL hybridization buffer and the bottles were rotated in the hybridization oven at 42°C overnight. Membranes were washed two times with 2× SSC/5% SDS for 15 min at 42°C and one time with 1× SSC/1% SDS for 15 min at 42°C.

Nucleic acids wash and immunological detection of the DIG-labeled probes were performed using the DIG Wash and Block Buffer Set (Roche, Cat. No. 11585762001) according to the manufacturer's protocols. Membranes were placed in a plastic container filled with 15 mL of the blocking solution for nonspecific binding sites blocking. After 30 min incubation at room temperature with rotation, 3 µL of the alkaline phosphatase-coupled anti-DIG antibody (Fab fragments, Roche, Cat No. 11093274910) were added to the buffer and incubation at room temperature on a belly-dancer was allowed for 30 more min. Next, membranes were washed three times 15 min with DIG-wash buffer and once with DIG-detection buffer for 5 min. For chemiluminescence visualization of the probe, 1 mL of CDP-Star Chemiluminescent Substrate disodium 2-chloro-5-(4-methoxyspiro[1,2-dioxetane-3,2'-[(5-chlorotricyclo[3.3.1.13.7]decan])-4-yl]-1-phenyl phosphate) (Roche, Cat. No. 11685627001) was added to 9 mL of DIG-detection buffer and membranes were then incubated with the substrate for 5 min. The chemiluminescent signal was detected with the iBright Imaging Systems.

## Ribosome profiling (Ribo-seq)

Pellet from 200 mL of *V. cholerae* at 0.25 OD 600 nm WT or mutant *Δtgt* in triplicates, with or without tobramycin were flash-frozen in liquid nitrogen and stored at –80°C. The polysomes were extracted with 200 µL of extraction buffer (20 mM Tris pH 8-150 mM Mg(CH$_3$COO)$_2$-100 mM NH4Cl, 5 mM CaCl$_2$-0,4% Triton X-100-1% Nonidet P40) added of 2× cocktail anti proteases Roche and 60U RNase Inhibitor Murine to the buffer, DNase I and glass beads (diameter <106 µm$^2$), vortexed during 30 min at 4°C. The supernatant of this crude extract was centrifugated 10 min at 21 krcf at +4°C. The absorbance was measured at 260 nm on 1 µL from 1/10 extract. After 1 hr of digestion at 25°C with 0.75 U

MNase/0.025 UA$_{260nm}$ of crude extract, the reaction was stopped by the addition of 3 µL 0.5 M EGTA pH 8. The monosomes generated by digestion were purified through a 24% sucrose cushion centrifuged 90 min at 110 krpm on a TLA110 rotor at +4°C. The monosomes pellet was rinsed with 200 µL of resuspension buffer (20 mM Tris-HCl pH 7.4-100 mM NH4Cl-15 mM Mg(CH$_3$COO)$_2$-5 mM CaCl$_2$) and then recovered with 100 µL. RNA were extracted by acid phenol at 65°C, CHCl$_3$ and precipitated by ethanol with 0.3 M CH$_3$COONa pH 5.2. Resuspended RNA was loaded on 17% polyacrylamide (19:1); 7 M urea in 1× TAE buffer at 100 V during 6 hr and stained with SYBRgold. RNA fragments corresponding to 28–34 nt were retrieved from gel and precipitated in ethanol with 0.3 M CH$_3$COONa pH 5.2 in presence of 100 µg glycogen. rRNA were depleted using MicrobExpress Bacterial mRNA Enrichment kit from Invitrogen. The supernatant containing the ribosome footprints were recovered and RNA were precipitated in ethanol in the presence of glycogen overnight at –20°C. The RNA concentration was measured by Quant-iT microRNA assay kit (Invitrogen). The RNA was dephosphorylated in 3' and then phosphorylated in 5' to generate cDNA libraries using the NebNext Small RNA Sample Prep kit with 3' sRNA Adapter (Illumina) according to the manufacturer's protocol with 12 cycles of PCR amplification in the last step followed by DNA purification with Monarch PCR DNA cleanup kit (NEB). Library molarity was measured with the Qubit DNAds HS assay kit from Invitrogen and the quality was analyzed using Bioanalyzer DNA Analysis kit (Agilent) and an equimolar pool of the 12 libraries was sequenced by the High-Throughput Sequencing Facility of I2BC with NextSeq 500/550 High Output Kit V2 (75 cycles) (Illumina) with 10% PhiX.

Sequencing data is available at GSE231087.

## Analysis of ribosome profiling data

RiboSeq analysis was performed using the RiboDoc package (*François et al., 2021*) for statistical analysis of differential gene expression (DEseq2). Sequencing reads are first trimmed to remove adaptors then aligned to the two *V. cholerae* chromosomes (NC_002505 and NC_002506). Reads aligned uniquely are used to perform the differential gene expression analysis. MNase shows significant sequence specificity at A and T (*Dingwall et al., 1981*). Due to this specificity and A-T biases in *V. cholerae* genome, ribosome profiling data exhibit a high level of noise that prevents the obtention of a resolution at the nucleotide level. The effect of ribosome stalling at TAT codons on total mRNA ribosome occupancy is likely highly variable, depending on the location of the TAT codon(s) within the CDS and the gene's expression level. We therefore interpreted genes in the 'Up' category as ones that mainly correspond to genes that are more translated because the impact of pausing at TAT codons is probably not strong enough.

## UV sensitivity measurements

Overnight cultures were diluted 1:100 in MH medium and grown with agitation at 37°C until an OD 600 nm of 0.5–0.7. Appropriate dilutions were then plated on MH agar. The proportion of growing CFUs after irradiation at 60 J over total population before irradiation was calculated, doing a ratio with total CFUs. Experiments were performed three to eight times.

## Quantification and statistical analysis

For comparisons between two groups, first an F-test was performed in order to determine whether variances are equal or different between comparisons. For comparisons with equal variance, Student's t-test was used. For comparisons with significantly different variances, we used Welch's t-test. For multiple comparisons, we used ANOVA to determine the statistical differences (p-value) between groups. **** means $p<0.0001$, *** means $p<0.001$, ** means $p<0.01$, * means $p<0.05$. For survival tests, data were first log-transformed in order to achieve normal distribution, and statistical tests were performed on these log-transformed data. The number of replicates for each experiment was $3<n<6$. Means and geometric means for logarithmic values were also calculated using GraphPad Prism.

## Bioinformatic analysis for whole genome codon bias determinations
### Data

Genomic data (fasta files containing CDS sequences and their translation, and GFF annotations) for *V. cholerae* (assembly ASM674v1) were downloaded from the NCBI FTP site (ftp://ftp.ncbi.nlm.nih.gov).

## Codon counting

For each gene, the codons were counted in the CDS sequence, assuming it to be in-frame. This step was performed using Python 3.8.3, with the help of the Mappy 2.20 (*Li, 2018*) and Pandas 1.2.4 (*McKinney, 2010*; *Reback et al., 2021*) libraries.

## Gene filtering

Genes whose CDS did not start with a valid start codon were excluded from further computations. A valid start codon is one among ATA, ATC, ATG, ATT, CTG, GTG, TTG, according to the genetic code for bacteria, archaea, and plastids (translation table 11 provided by the NCBI at ftp://ftp.ncbi.nlm.nih.gov/entrez/misc/data/gc.prt). Further computations were performed on 3590 genes that had a valid start codon.

## Codon usage bias computation

The global codon counts were computed for each codon by summing over the above selected genes. For each gene as well as for the global total, the codons were grouped by encoded amino acid. Within each group, the proportion of each codon was computed by dividing its count by the sum of the counts of the codons in the group. The codon usage bias for a given codon and a given gene was then computed by subtracting the corresponding proportion obtained from the global counts from the proportion obtained for this gene. Codon usage biases were then standardized by dividing each of the above difference by the standard deviation of these differences across all genes, resulting in standardized codon usage biases 'by amino acid' ('SCUB by aa' in short). All these computations were performed using the already mentioned Pandas 1.2.4 Python library.

## Associating genes to their preferred codon

For each codon group, genes were associated to the codon for which they had the highest 'SCUB by aa' value. This defined a series of gene clusters denoted using the 'aa_codon' pattern. For instance, 'Y_TAT' contains the genes for which TAT is the codon with the highest standardized usage bias among tyrosine codons.

## Extracting most positively biased genes from each cluster

Within each cluster, the distribution of 'SCUB by aa' values for each codon was represented using violin plots. Visual inspections of these violin plots revealed that in most cases, the distribution was multi-modal. An automated method was devised to further extract from a given cluster the genes corresponding to the sub-group with the highest 'SCUB by aa' for each codon. This was done by estimating a density distribution for 'SCUB by aa' values using a Gaussian kernel density estimate and finding a minimum in this distribution. The location of this minimum was used as a threshold above which genes were considered to belong to the most positively biased genes. This was done using the SciPy 1.7.0 (*Virtanen et al., 2020*) Python library. Violin plots were generated using the Matplotlib 3.4.2 (*Hunter, 2007*) and Seaborn 0.11.1 (*Waskom, 2021*) Python libraries.

# Acknowledgements

We thank Dr. Francesca Tuorto for sharing the protocol for boronated northern blots. Many thanks to Dr. Paola Arimondo for discussions about setting up the queuosine mass spectrometry experiments. We also thank, for RNA-seq experiments, E Turc, L Lemée, T Cokelaer, Biomics Platform, C2RT, Institut Pasteur, Paris, France, supported by France Génomique (ANR-10-INBS-09) and IBISA. We would like to acknowledge Valérie Bourguignon (IMoPA, UMR7365 CNRS-UL) for her help in validation of queuosine modification protocol. This research was funded by the Institut Pasteur, the Centre National de la Recherche Scientifique (CNRS-UMR 3525), ANR ModRNAntibio (ANR-21-CE35-0012), ANR-LabEx (ANR-10-LABX-62-IBEID), the Fondation pour la Recherche Médicale (FRM EQU202103012569), the Institut Pasteur grant PTR 245-19 and by the National Institute of General Medical Sciences (NIGMS) grant GM70641 to V dC-L. AB was funded by Institut Pasteur Roux-Cantarini fellowship. The authors acknowledge a DIM1Health 2019 grant from the Région Ile de France to the project EpiK for the LCMS equipment.

# Additional information

## Funding

| Funder | Grant reference number | Author |
|---|---|---|
| Centre National de la Recherche Scientifique | UMR3525 | Didier Mazel |
| Fondation pour la Recherche Médicale | 202103012569 | Louna Fruchard |
| Institut Pasteur | PTR 245-19 | Andre Carvalho |
| Agence Nationale de la Recherche | ANR-21-CE35-0012 | Anamaria Babosan Zeynep Baharoglu |
| Agence Nationale de la Recherche | ANR-10-LABX-62-IBEID | Didier Mazel |
| Agence Nationale de la Recherche | ANR-24-CE12-7224-01 | Louna Fruchard |
| National Institute of General Medical Sciences | GM70641 | Valérie de Crécy-Lagard |
| Institut Pasteur | Institut Pasteur Roux-Cantarini fellowship | Anamaria Babosan |

The funders had no role in study design, data collection and interpretation, or the decision to submit the work for publication.

## Author contributions

Louna Fruchard, Andre Carvalho, Investigation, Methodology; Anamaria Babosan, Manon Lang, Isabelle Hatin, Investigation; Blaise Li, Resources, Formal analysis, Methodology; Magalie Duchateau, Quentin Giai Gianetto, Mariette Matondo, Frederic Bonhomme, Resources, Investigation, Methodology; Hugo Arbes, Céline Fabret, Olivier Namy, Resources; Enora Corler, Formal analysis; Guillaume Sanchez, Virginie Marchand, Yuri Motorin, Validation, Investigation, Methodology; Valérie de Crécy-Lagard, Resources, Formal analysis, Writing – review and editing; Didier Mazel, Resources, Project administration; Zeynep Baharoglu, Conceptualization, Resources, Formal analysis, Supervision, Funding acquisition, Investigation, Methodology, Writing – original draft, Project administration, Writing – review and editing

## Author ORCIDs

Anamaria Babosan ⓘ https://orcid.org/0000-0003-3464-5491
Blaise Li ⓘ https://orcid.org/0000-0003-3080-1899
Magalie Duchateau ⓘ https://orcid.org/0000-0001-5475-3065
Mariette Matondo ⓘ https://orcid.org/0000-0003-3958-7710
Virginie Marchand ⓘ https://orcid.org/0000-0002-8537-1139
Olivier Namy ⓘ https://orcid.org/0000-0002-1143-5961
Valérie de Crécy-Lagard ⓘ https://orcid.org/0000-0002-9955-3785
Didier Mazel ⓘ https://orcid.org/0000-0001-6482-6002
Zeynep Baharoglu ⓘ https://orcid.org/0000-0003-3477-2685

Reviewer #1 (Public review): https://doi.org/10.7554/eLife.96317.3.sa1
Reviewer #2 (Public review): https://doi.org/10.7554/eLife.96317.3.sa2
Reviewer #3 (Public review): https://doi.org/10.7554/eLife.96317.3.sa3
Author response https://doi.org/10.7554/eLife.96317.3.sa4

# Additional files

## Supplementary files
Supplementary file 1. Table of differentially abundant proteins identified in proteomics.

Supplementary file 2. Table of ribosome profiling data.

Supplementary file 3. RNA-seq *V. cholerae* WT/Δtgt, in MH and TOB.

Supplementary file 4. Table of primers, plasmids, and strains.

MDAR checklist

## Data availability

For ribosome profiling, sequencing data is available at GSE231087. The mass spectrometry proteomics data have been deposited to the ProteomeXchange Consortium via the PRIDE partner repository with the dataset identifier PXD035297. The data for RNA-seq has been submitted in the GenBank repository under the project number GSE214520. Data are available for whole genome codon usage of V. cholerae in excel sheet and V. cholerae codon usage biased gene lists at Zenodo public repository. All codes to perform these analyses were implemented in the form of Python scripts, Jupyter notebooks (*Kluyver et al., 2016*), and Snakemake (*Mölder et al., 2021*) workflows and are available in a git repository (copy archived at *Li, 2022*).

The following datasets were generated:

| Author(s) | Year | Dataset title | Dataset URL | Database and Identifier |
|---|---|---|---|---|
| Baharoglu Z | 2022 | *Vibrio cholerae* codon biased genes list | https://zenodo.org/records/6875293 | Zenodo, 10.5281/zenodo.6875293 |
| Duchateau M, Baharoglu Z | 2024 | Queuosine modification of tRNA-Tyr elicits translational reprogramming and enhances growth of *Vibrio cholerae* with aminoglycosides | http://www.ebi.ac.uk/pride/archive/projects/PXD035297 | PRIDE, PXD035297 |
| Fruchard L, Babosan A, Carvalho A, Lang M, Li B, Duchateau M, Giai-Gianetto Q, Matondo M, Bonhomme F, Fabret C, Namy O, Crécy-Lagard Mazel D, Baharoglu Z | 2022 | *V. cholerae* transcriptome or deleted for the tgt gene in MH, supplemented or not with tobramycin or ciprofloxacin | https://www.ncbi.nlm.nih.gov/geo/query/acc.cgi?acc=GSE214520 | NCBI Gene Expression Omnibus, GSE214520 |
| Fruchard L, Babosan A, Carvalho A, Lang M, Li B, Duchateau M, Giai-Gianetto Q, Matondo M, Bonhomme F, Hatin I, Arbes H, Fabret C, Marchand V, Motorine I, Namy O, Crécy-Lagard deV, Mazel D, Baharoglu Z | 2024 | Aminoglycoside tolerance in Vibrio cholerae engages translational reprogramming associated to queuosine tRNA modification | https://www.ncbi.nlm.nih.gov/geo/query/acc.cgi?acc=GSE231087 | NCBI Gene Expression Omnibus, GSE231087 |

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
