## [Editor Report · eLife Assessment]

This study investigates the role of queuosine (Q) tRNA modification in aminoglycoside tolerance in *Vibrio cholerae* and presents **convincing** evidence to conclude that Q is essential for the efficient translation of TAT codons, although this depends on the context. The absence of Q reduces aminoglycoside tolerance potentially by reprogramming the translation of an oxidative stress response gene, rxsA. Overall, the findings point to an **important** mechanism whereby changes in Q modification levels control the decoding of mRNAs enriched in TAT codons under antibiotic stress.

---

## [Referee Report · Reviewer #1 (Public review)]

Summary of the work:

In this work Fruchard et. al. study the enzyme Tgt and how it modifies guanine in tRNAs to queuosine (Q), essential for *Vibrio cholerae*'s growth under aminoglycoside stress. Q's role in codon decoding efficiency and its proteomic effects during antibiotic exposure is examined, revealing Q modification impacts tyrosine codon decoding and influences RsxA translation, affecting the SoxR oxidative stress response. The research proposes Q modification's regulation under environmental cues reprograms the translation of genes with tyrosine codon bias, including DNA repair factors, crucial for bacterial antibiotic response.

The experiments are well-designed and conducted and the conclusions, for the most part, are well-supported by the data.

Comments on revisions:

The authors have answered my queries.

---

## [Referee Report · Reviewer #2 (Public review)]

Fruchard et al. investigate the role of the queuosine (Q) modification of the tRNA (Q-tRNA) in the human pathogen *Vibrio cholerae*. First, the authors state that the absence of Q-modified tRNAs (tgt mutant) increases the translation of TAT codons and proteins with a high TAT codon bias. Second, the absence of Q increases rsxA translation, because rsxA gene has a high TAT codon bias. Third, increased RsxA in the absence of Q inhibits SoxR response, reducing resistance towards the antibiotic tobramycin (TOB). Authors also predict in silico which genes harbor a higher TAT bias and found that among them are some involved in DNA repair, experimentally observing that a tgt mutant is more resistant to UV than the wt strain. It is worth noting that authors employ a wide variety of techniques, both experimental and bioinformatics.

The authors have satisfactorily responded to most of the comments that needed clarification. Particularly interesting was the addition of the new results section "Q modification impacts decoding fidelity in *V. cholerae*", after the suggestion to explore the role of Q in prevention of stop codon readthrough. Although it is not a major problem, since the article is very complete and interesting, the interpretation of the results of RiboSeq data carried out in this work remains controversial. This technique, at least when it has been used in eukaryotes to investigate whether there is a bias in the translation of certain codons affected by Q (Tuorto et al., EMBO J. 2018; doi: 10.15252/embj.201899777), has been interpreted as meaning that ribosomes spend less time in the optimal codons and therefore there is an increase in occupancy at codons where translation slows down. On the other hand, it has been observed that "in ribosome profiling experiments conducted without cycloheximide pretreatment, there is a clear inverse relationship between tRNA abundance and ribosome occupancy, showing that ribosomes spend less time at optimal codons and specifically this has been observed in experiments in which a translation inhibitor such as cycloheximide is not used (see review: Hanson G & Coller J. Nat Rev Mol Cell Biol. doi: 10.1038/nrm.2017.91, and experiments in yeast: Hussmann JA et al. PLoS Genet. doi: 10.1371/journal.pgen.1005732). On the other hand, we believe that the comparison between RiboSeq and proteomic data would be interesting to check whether this interpretation of the RiboSeq data is correct. It should not be a problem that the proteomics data could be incomplete, it would just be a more limited study. If the correct interpretation of the RiboSeq results is as proposed by the authors, a correlation should be observed between the abundance of TAT-enriched RNA fragments and the most abundant proteins. Therefore, it would be interesting to perform this comparison and see if significant results are obtained that help to understand the correct interpretation of the RiboSeq experiments.

---

## [Referee Report · Reviewer #3 (Public review)]

Summary:

In this manuscript the authors begin with the interesting phenotype of sub inhibitory concentrations of the aminoglycoside tobramycin proving toxic to a knockout of the tRNA-guanine transglycosylase (Tgt) of the important human pathogen, *Vibrio cholerae*. Tgt is important for incorporating queuosine (Q) in place of guanosine at the wobble position of GUN codons. The authors go on to define a mechanism of action where environmental stressors control expression of tgt to control translational decoding of particularly tyrosine codons, skewing the balance from TAC towards TAT decoding in the absence of the enzyme. The authors use advanced proteomics and ribosome profiling to reveal that the loss of tgt results in increased translation of proteins like RsxA and a cohort of DNA repair factors, whose genes harbor an excess of TAT codons in many cases. These findings are bolstered by a series of molecular reporters, mass spectrometry, and tRNA overexpression strains to provide support for a model where Tgt serves as a molecular pivot point to reprogram translational output in response to stress. The manuscript therefore improves our understanding of the phenotype of focus and will prove useful for the field in our understanding of Modification Tunable Transcripts.

Strengths:

The manuscript has many strengths. The authors use a variety of strains, assays, and advanced techniques to discover a mechanism of action for Tgt in mediating tolerance to sub inhibitory concentrations of tobramycin. They observe a clear phenotype for a tRNA modification in facilitating reprogramming of the translational response, and the manuscript certainly has value in defining how microbes tolerate antibiotics.

Weaknesses:

The conclusions of the manuscript are mostly very well-supported by the data, but a few experimental directions remain inconclusive. The finding linking Tgt and UV damage susceptibility is one example where the phenotype is striking, but the mechanism remains somewhat unclear. Future work in this direction will likely be required to fully understand how Tgt influences the repair of DNA after UV.

---

## [Author Response]

The following is the authors’ response to the original reviews.

**Public Reviews:**

**Reviewer #1 (Public Review):**
Summary of the work: In this work, Fruchard et. al. study the enzyme Tgt and how it modifies guanine in tRNAs to queuosine (Q), essential for *Vibrio cholerae*'s growth under aminoglycoside stress. Q's role in codon decoding efficiency and its proteomic effects during antibiotic exposure is examined, revealing Q modification impacts tyrosine codon decoding and influences RsxA translation, affecting the SoxR oxidative stress response. The research proposes Q modification's regulation under environmental cues reprograms the translation of genes with tyrosine codon bias, including DNA repair factors, crucial for bacterial antibiotic response.The experiments are well-designed and conducted and the conclusions, for the most part, are well supported by the data. However, a few clarifications will significantly strengthen the manuscript.

Thank you.

Major:Figure S4 A-D. These growth curves are important data and should be presented in the main figures. Moreover, given that it is not possible to make a rsxA mutant, I wonder if it would be possible to connect rsx and tgt using the following experiment: expression of tgt results in resistance to TOB (in B), while expression of only rsx lower resistance to TOB (in D). Then simultaneous overexpression of both tgt/rsx in the WT strain should have either no effect on TOB resistance or increased resistance, relative to the WT. Perhaps the authors have done this, and if so, the data should be included as it will significantly strengthen their model.

We thank the reviewer for this suggestion, we have tried to overexpress both tgt and rsxA simultaneously. However, this appears to be toxic as cells form small colonies and cannot grow well in liquid. We think that the presence of 2 plasmids and corresponding selection antibiotics amplify the toxicity of overexpressing *rsxA*, and even *tgt*. In fact, it can be seen that *tgt* overexpression in WT is already slightly deleterious, in the absence of tobramycin (figure 1B).

Figure S4 - Is there a rationale for why it is possible to make rsx mutants in *E. coli*, but not in V. cholerae? For example, does E. coli have a second gene/protein that is redundant in function to rsxA, while V. cholerae does not? I think your data hint at this, since in the right panel growth data, your double mutant does not fully rescue back to rsx single mutant levels, suggesting another factor in tgt mutant also acts to lower resistance to TOB. If so, perhaps a line or two in text will be helpful for readers.

This point raised by the referee is an interesting one that we have also asked ourselves at multiple occasions. In fact, the Rsx operon is linked with oxidative stress and respiration. *Vibrio cholerae* and E. coli show differences on genes involved in these pathways. V. cholerae lacks the cyo/nuo respiratory complex genes, and does not encode a Suf operon. Moreover, deletion of the anaerobic respiration Frd pathway leads to strong decrease of V. cholerae growth even in aerobic conditions. (10.1128/spectrum.01730-23). We have previously also generally seen differences between the 2 species in response to stress (10.1128/AAC.01549-10) and the way they deal with ROS (10.1371/journal.pgen.1003421). Therefore, we think that the fact that rsx is essential in *V. cholerae* and not *E. coli* could either be due to the presence of an additional redundant pathway in *E. coli* as suggested by the referee, or to more general differences in respiration and treatment of ROS. We thank the referee for highlighting this and we have now included a comment about this in the manuscript.

- For growth curves in Figure 2 and relative comparisons like in Figure 5D and Figure S4 (and others in the paper), statistics and error bars, along with replicate information should be provided.

We had mentioned this in the methods section, we have now added the specific information also on figure legends.

- Figure 6A - Is the transcript fold change in linear or log? If linear, then tgt expression should not be classified as being upregulated in TOB. It is barely up by ~2-fold with TOB- 0.6....which is a mild phenotype, at best.

We think that 2-fold change of tgt expression can be sufficient to lead to changes in tRNA modification levels. We agree that this is a mild induction, we have thus changed “increase” to “mildly increase” in the results.

- Line 779- 780: "This indicates that sub-MIC TOB possibly induces tgt expression through the stringent response activation." To me, the data presented in this figure, do not support this statement. The experiment is indirect.

We agree, we rephrased: “Tobramycin may induces *tgt* expression through stringent response activation or through an independent pathway. “

- Figure 3B and D. - These samples only have tobramycin, correct? The legend says both carbenicillin and tobramycin.

The legend is correct, samples also have carbenicillin because we are testing here the growth with 2 synonymous beta-lactamase genes in presence of beta-lactams.

- Figure 5. The color schemes in bars do not match up with the color scheme in cartoons below panels B and C. That makes it confusing to read. Please fix.

Fixed.

- A lot of abbreviations have been used. This makes reading a bit cumbersome. Ideally, less abbreviations will be used.

Fixed.

**Reviewer #2 (Public Review):**
Fruchard et al. investigate the role of the queuosine (Q) modification of the tRNA (Q-tRNA) in the human pathogen *Vibrio cholerae*. First, the authors state that the absence of Q-modified tRNAs (tgt mutant) increases the translation of TAT codons and proteins with a high TAT codon bias. Second, the absence of Q increases rsxA translation, because rsxA gene has a high TAT codon bias. Third, increased RsxA in the absence of Q inhibits SoxR response, reducing resistance towards the antibiotic tobramycin (TOB). Authors also predict in silico which genes harbor a higher TAT bias and found that among them are some involved in DNA repair, experimentally observing that a tgt mutant is more resistant to UV than the wt strain. It is worth noting that authors employ a wide variety of techniques, both experimental and bioinformatic. However, some aspects of the work need to be clarified or reevaluated.(1) The statement that the absence of Q increases the translation of TAT codons and proteins encoded by TAT-enriched genes presents the following problems that should be addressed:(1.1) The increase in TAT codon translation in the absence of Q is not supported by proteomics, since there was no detected statistical difference for TAT codon usage in proteins differentially expressed. Furthermore, there are some problems regarding the statistics of proteomics. Some proteins shown in Table S1 have adjusted p-values higher than their pvalues, which makes no sense. Maybe there is a mistake in the adjusted p-value calculation.

We appreciate the reviewer’s thorough examination of our findings. In our study, we employed an adaptive Benjamini-Hochberg (BH) procedure to control the false discovery rate in our list of selected proteins, as explained in the Data Analysis part of the Proteomics MS and analysis part of our material and methods. The classical BH procedure (10.1111/j.2517-6161.1995.tb02031.x) calculates the 𝑚×𝑝(𝑗) adjusted p-value for the i-th ranked p-value as min (1,minj≥im×p(j)j)) where 𝑝(𝑗) is the j-th ranked pvalue and 𝑚 is the number of tests (e.g. number of proteins) (see 10.1021/acs.jproteome.7b00170 for details). Since m/j > 1 and 𝑝(𝑗) > 𝑝(𝑖) for 𝑗≥𝑚, it follows that m×p(j)j(j)(i) for 𝑗≥*i*, resulting in adjusted p-values being higher or equal than the original p-values. Therefore, contrary to the reviewer's comment, it is a mathematical property that the adjusted p-value is greater than the original p-value when using the classical Benjamini-Hochberg procedure.

However, we want to underline that we used an « adaptive » BH procedure, which calculates the adjusted p-value for the i-th ranked p-value as min (1,minj≥im×π0×p(j)j)), where 𝜋0 is an estimate of the proportion of true null hypotheses see 10.1021/acs.jproteome.7b00170 for details. Indeed, the classical BH procedure makes the assumption that 𝜋0 = 1, which is a strong assumption in MS-based proteomics context. Consequently, the mathematical property that the adjusted p-value is greater than the original p-value does not always hold true in our approach (that depends also on the 𝜋0 parameter).

In addition, it is not common to assume that proteins that are quantitatively present in one condition and absent in another are differentially abundant proteins. Proteomics data software typically addresses this issue and applies some corrections. It would be advisable to review that.

We thank the reviewer for highlighting this point. Indeed, some software impute a random small value to replace missing values and then produces statistics based on this imputed data (10.1038/nmeth.3901). However, the validity and relevance of generating statistics in the absence of actual data is questionable.

There are no universally accepted guidelines for handling this situation, and we believe it is more logical to set these values aside as potential interesting proteins. It is well-established that intensity values are often missing due to the detection limits of the spectrometer, suggesting that the missing values observed in several replicates of a condition are actually due to low values (see 10.1093/bioinformatics/btp362 and 10.1093/bioinformatics/bts193 for instance). It is thus logical to consider the associated proteins as potentially differentially abundant when comparing their complete absence in all replicates of one condition to their presence in several replicates of another condition.

(1.2) Problems with the interpretation of Ribo-seq data (Figure 4D). On the one hand, the Ribo-seq data should be corrected (normalized) with the RNA-seq data in each of the conditions to obtain ribosome profiling data, since some genes could have more transcription in some of the conditions studied. In other articles in which this technique is used (such as in Tuorto et al., EMBO J. 2018; doi: 10.15252/embj.201899777), it is interpreted that those positions in which the ribosome moves most slowly and therefore less efficiently translated, are the most abundant. Assuming this interpretation, according to the hypothesis proposed in this work, the fragments enriched in TAT codons should have been less abundant in the absence of Q-tRNA (tgt mutant) in the Rib-seq experiment. However, what is observed is that TAT-enriched fragments are more abundant in the tgt mutant, and yet the Ribo-seq results are interpreted as RNA-seq, stating that this is because the genes corresponding to those sequences have greater expression in the absence of Q.

As recommended by the reviewer, we normalized the RiboSeq data with the RNAseq data to account for potential RNA variations. The updated Figure 4 demonstrates that this normalization does not alter our findings, confirming that variations at the RNAseq level do not contradict changes at the translational level.

The reviewer's observation that pauses at TAT codons would lead to ribosome accumulation and subsequent categorization as "up" genes is accurate. We must emphasize, however, that this category of “up genes” is probably quite diverse. The effect of ribosome stalling at TAT codons on total mRNA ribosome occupancy is likely highly variable, depending on the location of the TAT codon(s) within the CDS and the gene's expression level. We therefore think that genes in the "Up" category mainly correspond to genes that are more translated because the impact of pausing at TAT codons is probably not strong enough. Note that unlike what is usually done in bacterial riboseq experiments, we did not use any antibiotics to artificially freeze the ribosomes.

On the other hand, it would be interesting to calculate the mean of the protein levels encoded by the transcripts with high and low ribosome profiling data.

While this is a common request, we believe that comparing RiboSeq and proteomics data is not particularly informative. RiboSeq data directly measures translation, while proteomics provides information about protein abundance at steady state, reflecting the balance between protein synthesis and degradation. Furthermore, the number of proteins detectable by mass spectrometry is significantly smaller than the number of genes quantified by RiboSeq. Given these factors, there is often a low correlation between translation and protein abundance, making a direct comparison less relevant

(1.3) This statement is contrary to most previously reported studies on this topic in eukaryotes and bacteria, in which ribosome profiling experiments, among others, indicate that translation of TAT codons is slower (or unaffected) than translation of the TAC codons, and the same phenomenon is observed for the rest of the NAC/T codons. This is completely opposed to the results showed in Figure 4. However, the results of these studies are either not mentioned or not discussed in this work. Some examples of articles that should be discussed in this work:- "Queuosine-modified tRNAs confer nutritional control of protein translation" (Tuorto et al., 2018; 10.15252/embj.201899777)- "Preferential import of queuosine-modified tRNAs into *Trypanosoma* brucei mitochondrion is critical for organellar protein synthesis" Kulkarni et al., 2021; doi:10.1093/nar/gkab567.- "Queuosine-tRNA promotes sex-dependent learning and memory formation by maintaining codonbiased translation elongation speed" (Cirzi et al., 2023; 10.15252/embj.2022112507)- "Glycosylated queuosines in tRNAs optimize translational rate and post-embryonic growth" (Zhao et al., 2023; 10.1016/j.cell.2023.10.026)- "tRNA queuosine modification is involved in biofilm formation and virulence in bacteria" (Diaz-Rullo and Gonzalez-Pastor, 2023; doi: 10.1093/nar/gkad667). In this work, the authors indicate that QtRNA increases NAT codon translation in most bacterial species. Could the regulation of TAT codonenriched proteins by Q-tRNAs in *V. cholerae* an exception? In addition, authors use a bioinformatic method to identify genes enriched in NAT codons similar to the one used in this work, and to find in which biological process are involved the genes whose expression is affected by Q-tRNAs (as discussed for the phenotype of UV resistance). It will be worth discussing all of this.

Thank you for detailed suggestions, we agree that this discussion was missing and this comment gives us a chance to address that in the revised version of the manuscript.

About the references above suggested by the referee, 4 of these papers were not mentioned in our manuscript, these were published while our manuscript was previously in review and we realize we have not cited them in the latest version of our manuscript. We thank the referee for highlighting this. We have now included a discussion about this.

We included the following in the discussion:

“However, the opposite codon preference was shown in *E. coli* {Diaz-Rullo, 2023 #1888}. In eukaryotes also, several recent studies indicate slower translation of U-ending codons in the absence of Q34 {Cirzi, 2023 #1887;Kulkarni, 2021 #1886;Tuorto, 2018 #1268}. It’s important to note here, that in *V. cholerae* ∆tgt, increased decoding of U-ending codons is observed only with tyrosine, and not with the other three NAC/U codons (Histidine, Aspartate, Asparagine). This is interesting because it suggests that what we observe with tyrosine may not adhere to a general rule about the decoding efficiency of U- or C-ending codons, but instead seems to be specific to Tyr tRNAs, at least in the context of *V. cholerae*. Exceptions may also exist in other organisms. For example, in human cells, queuosine increases efficiency of decoding for U- ending codons and slows decoding of C- ending codons except for AAC {Zhao, 2023 #1889}. In this case, the exception is for tRNA Asparagine. Moreover, in mammalian cells {Tuorto, 2018 #1268}, ribosome pausing at U-ending codons is strongly seen for Asp, His and Asn, but less with Tyr. In Trypanosoma {Kulkarni, 2021 #1886}, reporters with a combination of the 4 NAC/NAU codons for Asp, Asn, Tyr, His have been tested, showing slow translation at U- ending version of the reporter in the absence of Q, but the effect on individual codons (e.g. Tyr only) is not tested. In mice {Cirzi, 2023 #1887}, ribosome slowdown is seen for the Asn, Asp, His U-ending codons but not for the Tyr U-ending codon. In summary, Q generally increases efficiency of U- ending codons in multiple organisms, but there appears to be additional unknown parameters which affect tyrosine UAU decoding, at least in *V. cholerae*. Additional factors such as mRNA secondary structures or mistranslation may also contribute to the better translation of UAU versions of tested genes. Mistranslation could be an important factor. If codon decoding fidelity impacts decoding speed, then mistranslation could also contribute to decoding efficiency of Tyr UAU/UAC codons and proteome composition.”

(1.4) It is proposed that the stress produced by the TOB antibiotic causes greater translation of genes enriched in TAT codons.

Actually, it’s the opposite because in presence of TOB, in the wt, tgt would be induced leading to more Q on tRNA-Tyr and less translation of TAT.

On the one hand, it is shown that the GFP-TAT version (gene enriched in TAT codons) and the RsxATAT-GFP protein (native gene naturally enriched in TAT) are expressed more, compared to their versions enriched in TAC in a tgt mutant than in a wt, in the presence of TBO (Fig. 5C).

Figure 5C shows relative fluorescence, ie changes of fluorescence in delta-tgt compared to WT. So it’s not necessarily more expressed but “more increased”

However, in the absence of TOB, and in a wt context, although the two versions of GFP have a similar expression level (Fig. 3SD), the same does not occur with RsxA, whose RsxA-TAT form (the native one) is expressed significantly more than the RsxA-TAC version (Fig. 3SA). How can it be explained that in a wt context, in which there are also tRNA Q-modification, a gene naturally enriched in TAT is translated better than the same gene enriched in TAC?

We thank the referee for this question based on careful assessment of our data. We agree, there appears to be significantly more RsxA-TAT in WT than RsxA-TAC. This could be due to other effects such as secondary structure formation on mRNA when the wt RsxA is recoded with TAC codons. This does not hinder the conclusion that the translation of the TAT version is increased in delta-tgt compared to WT.

It would be expected that in the presence of Q-tRNAs the two versions would be translated equally (as happens with GFP) or even the TAT version would be less translated. On the other hand, in the presence of TOB the fluorescence of WT GFP(TAT) is higher than the fluorescence of WT GFP(TAC) (Figure S3E) (mean fluorescence data for RsxA-GFP version in the presence of TOB is not shown). These results may indicate that the apparent better translation of TAT versions could be due to indirect effects rather from TAT codon translation.

This is now mentioned in the manuscript:

“We cannot exclude, however, that additional factors such as mRNA secondary structures also contributes to the better translation of UAU versions of tested genes. “

(2) Another problem is related to the already known role of Q in prevention of stop codon readthrough, which is not discuss at all in the work. In the absence of Q, stop codon readthrough is increased. In addition, it is known that aminoglycosides (such as tobramycin) also increase stop codon readthrough ("Stop codon context influences genome-wide stimulation of termination codon readthrough by aminoglycosides"; Wanger and Green, 2023; 10.7554/eLife.52611). Absence of Q and presence of aminoglycosides can be synergic, producing devastating increases in stop codon readthrough and a large alteration of global gene expression. All of these needs to be discussed in the work. Moreover, it is known that stop codon readthrough can alter gene expression and mRNA sequence context all influence the likelihood of stop codon readthrough. Thus, this process could also affect to the expression of recoded GFP and RsxA versions.

We included the following in the revised version of the manuscript (results):

“Q modification impacts decoding fidelity in *V. cholerae*.

To test whether a defect in Q34 modification influences the fidelity of translation in the presence and absence of tobramycin, previously developed reporter tools were used (Fabret & Namy, 2021), to measure stop codons readthrough in *V. cholerae* ∆tgt and wild-type strains. The system consists of vectors containing readthrough promoting signals inserted between the lacZ and luc sequences, encoding β-galactosidase and luciferase, respectively. Luciferase activity reflects the readthrough efficiency, while β-galactosidase activity serves as an internal control of expression level, integrating a number of possible sources of variability (plasmid copy number, transcriptional activity, mRNA stability, and translation rate). We found increased readthrough at stop codons UAA and to a lesser extent at UAG for ∆tgt, and this increase was amplified for UAG in presence of tobramycin (Fig. S2, stop readthrough). In the case of UAA, tobramycin appears to decrease readthrough, this may be artefactual, due to the toxic effect of tobramycin on ∆tgt.

Mistranslation at specific codons can also impact protein synthesis. To further investigate mistranslation levels by tRNATyr in WT and ∆tgt, we designed a set of gfp mutants where the codon for the catalytic tyrosine required for fluorescence (TAT at position 66) was substituted by nearcognate codons (Fig. S2). Results suggest that in this sequence context, particularly in the presence of tobramycin, non-modified tRNATyr mistakenly decodes Asp GAC, His CAC and also Ser UCC, Ala GCU, Gly GGU, Leu CUU and Val GUC codons, suggesting that Q34 increases the fidelity of tRNATyr.

In parallel, we replaced Tyr103 of the β-lactamase described above, with Asp codons GAT or GAC. The expression of the resulting mutant β-lactamase is expected to yield a carbenicillin sensitive phenotype. In this system, increased tyrosine misincorporation (more mistakes) by tRNATyr at the mutated Asp codon, will lead to increased synthesis of active β-lactamase, which can be evaluated by carbenicillin tolerance tests. As such, amino-acid misincorporation leads here to phenotypic (transient) tolerance, while genetic reversion mutations result in resistance (growth on carbenicillin). The rationale is summarized in Fig. 3C. When the Tyr103 codon was replaced with either Asp codons, we observe increased β-lactamase tolerance (Fig. 3D, left), suggesting increased misincorporation of tyrosine by tRNATyr at Asp codons in the absence of Q, again suggesting that Q34 prevents misdecoding of Asp codons by tRNATyr.

In order to test any effect on an additional tRNA modified by Tgt, namely tRNAAsp, we mutated the Asp129 (GAT) codon of the β-lactamase. When Asp129 was mutated to Tyr TAT (Fig. 3D, right), we observe reduced tolerance in ∆tgt, but not when it was mutated to Tyr TAC, suggesting less misincorporation of aspartate by tRNAAsp at the Tyr UAU codon in the absence of Q. In summary, absence of Q34 increases misdecoding by tRNATyr at Asp codons, but decreases misdecoding by tRNAAsp at Tyr UAU.

This supports the fact that tRNA Q34 modification is involved in translation fidelity during antibiotic stress, and that the effects can be different on different tRNAs, e.g. tRNATyr and tRNAAsp tested here.”

Added figures: Figure S2, Figure 3CD

(3) The statement about that the TOB resistance depends on RsxA translation, which is related to the presence of Q, also presents some problems:(3.1) It is observed that the absence of tgt produces a growth defect in *V. cholerae* when exposed to TOB (Figure 1A), and it is stated that this is mediated by an increase in the translation of RsxA, because its gene is TAT enriched. However, in Figure S4F, it is shown that the same phenotype is observed in *E. coli*, but its rsxA gene is not enriched in TAT codons. Therefore, the growth defect observed in the tgt mutant in the presence of TOB may not be due to the increase in the translation of TAT codons of the rsxA gene in the absence of Q. This phenotype is very interesting, but it may be related to another molecular process regulated by Q. Maybe the role of Q in preventing stop codon readthrough is important in this process, reducing cellular stress in the presence of TOB and growing better.

FigS4F (now figure 5D) shows that rsxA can be toxic during growth in presence of tobramycin, but it does not show that rsxA translation is increased in *E. coli* in delta-tgt. However, we agree with the referee that there are probably additional processes regulated by Q which are also involved in the response to TOB stress. We already had mentioned this briefly in the discussion (“Note that, our results do not exclude the involvement of additional Q-regulated MoTTs in the response to sub-MIC TOB, since Q modification leads to reprogramming of the whole proteome.”), we further discussed it as follows:

“As a consequence, transcripts with tyrosine codon usage bias are differentially translated. One such transcript codes for RsxA, an anti-SoxR factor. SoxR controls a regulon involved in oxidative stress response and sub-MIC aminoglycosides trigger oxidative stress in *V. cholerae*{Baharoglu, 2013 #720}, pointing to an involvement of oxidative stress response in the response to sub-MIC tobramycin stress.

A link between Q34 and oxidative stress has also been previously found in eukaryotic organisms {Nagaraja, 2021 #1466}. Note that our results do not exclude the involvement of additional Qregulated translation of other transcripts in the response to tobramycin. Q34 modification leads to reprogramming of the whole proteome, not only for other transcripts with codon usage bias, but also through an impact on the levels of stop codon readthrough and mistranslation at specific codons, as supported by our data.”

(3.2) All experiments related to the effect of Q on the translation of TAT codons have been performed with the tgt mutant strain. Considering that the authors have a pSEVA-tgt plasmid to overexpress this gene, they would have to show whether tgt overexpression in a wt strain produces a decrease in the translation of proteins encoded by TAT-enriched genes such as RsxA. This experiment would allow them to conclude that Q reduces RsxA levels, increasing resistance to TOB.

We agree that this would be interesting to test, however, as it can be seen in figure 1B, delta-tgt pSEVAtgt (complemented strain) grows better than WT pSEVA-tgt (tgt overexpression). In fact, overexpression of tgt negatively impacts cell growth and yield smaller colonies, especially when cells carry a second plasmid (e.g with gfp constructs). We have also seen this with other RNA modification gene overexpressions in the lab (unpublished). We believe that the expression of tgt is tuned and since overexpression affects fitness, it is generally difficult to conduct experiments with overexpression plasmid for RNA modifications. Nevertheless, we have done the experiment (with slow growing bacteria) and when we normalize expression of gfp in the presence of tgt overexpressing plasmid to the condition with no plasmid, we see little (1.5 fold) or no effect of tgt overexpression on fluorescence (see graph below). This is probably due to a toxic effect of ooverexpression and we do not believe these results are biologically relevant.

**Author response image 1. sa4fig1:** 

(3.3) On the other hand, Fig. 1B shows that when the wt and tgt strains compete, both overexpressing tgt, the tgt mutant strain grows better in the presence of TOB. This result is not very well understood, since according to the hypothesis proposed, the absence of modification by Q of the tRNA would increase the translation of genes enriched in TAT, therefore, a strain with a higher proportion of Q-modified tRNAs as in the case of the wt strain overexpressing tgt would express the rsxA gene less than the tgt strain overexpressing tgt and would therefore grow better in the presence of TOB. For all these reasons, it would be necessary to evaluate the effect of tgt overexpression on the translation of RsxA.

See our answer above about negative effect of *tgt* overexpression.

(3.4) According to Figure 1I, the overexpression of tRNA-Tyr(GUA) caused a better growth of tgt mutant in comparison to WT. If the growth defect observed in tgt mutant in the presence of TOB is due to a better translation of the TAT codons of rsxA gene, the overexpression of tRNA-Tyr(GUA) in the tgt mutant should have resulted in even better RsxA translation a worse growth, but not the opposite result.

We agree, we think that rsxA is not the only factor responsible for growth defect of tgt in presence of TOB (as now further discussed in the discussion). Overexpression of tRNAtyr possibly changes the equilibrium between the decoding of TAC vs TAT and may restore translation of TAC enriched genes. As also suggested by rev3, we have measured decoding reporters for TAT/TAC while overexpressing tTNA-tyr. This is now added to the results in fig S2C and the following:

“We also tested decoding reporters for TAT/TAC in WT and ∆tgt overexpressing tRNATyr in trans (Fig. S1C). The presence of the plasmid (empty p0) amplified differences between the two strains with decreased decoding of TAC (and increased TAT, as expected) in ∆tgt compared to WT. Overexpression of tRNATyrGUA did not significantly impact decoding of TAT and increased decoding of TAC, as expected. Since overexpression of tRNATyrGUA rescues ∆tgt in tobramycin (Fig. 1I) and facilitates TAC decoding, this suggests that issues with TAC codon decoding contribute to the fitness defect observed in ∆tgt upon growth with tobramycin. Overexpression of tRNATyrAUA increased decoding of TAT in WT but did not change it in ∆tgt where it is already high. Unexpectedly, overexpression of tRNATyrAUA also increased decoding of TAC in WT. Thus, overexpression of tRNATyrAUA possibly changes the equilibrium between the decoding of TAC vs TAT and may restore translation of TAC enriched transcripts.”

Added figure: figure S1C

(4) It cannot be stated that DNA repair is more efficient in the tgt mutant of *V. cholerae*, as indicated in the text of the article and in Fig 7. The authors only observe that the tgt mutant is more resistant to UV radiation and it is suggested that the reason may be TAT bias of DNA repair genes. To validate the hypothesis that UV resistance is increased because DNA repair genes are TAT biased, it would be necessary to check if DNA repair is affected by Q. UV not only produces DNA damage, but also oxidative stress. Therefore, maybe this phenotype is due to the increase in proteins related to oxidative stress controlled by RsxA, such as the superoxide dismutase encoded by sodA. It is also stated that these repair genes were found up for the tgt mutant in the Ribo-seq data, with unchanged transcription levels. Again, it is necessary to clarify this interpretation of the Ribo-seq data, since the fact that they are more represented in a tgt mutant perhaps means that translation is slower in those transcripts. Has it been observed in proteomics (wt vs tgt in the absence of TOB) whether these proteins involved in repair are more expressed in a tgt mutant?

We agree that our results do not directly show that DNA repair is more efficient, but that delta-tgt responds better to UV. This has been modified in the manuscript. About oxidative stress, we did not see a better or worse response to H202 of delta-tgt. Moreover, since we see better response of deltatgt to UV only in V. cholerae and not in *E. coli*, we did not favor the hypothesesi of response to stressox. In proteomics, we do not detect changes for DNA repair genes except for RuvA which is more abundant in delta-tgt. We have toned down the statement about DNA repair in the paper.

(5) The authors demonstrate that in *E. coli* the tgt mutant does not show greater resistance to UV radiation (Fig. 7D), unlike what happens in *V. cholerae*. It should be discussed that in previous works it has been observed that overexpression in *E. coli* of the tgt gene or the queF gene (Q biosynthesis) is involved in greater resistance to UV radiation (Morgante et al., Environ Microbiol, 2015 doi: 10.1111/1462-2920.12505; and Díaz-Rullo et al., Front Microbiol. 2021 doi: 10.3389/fmicb.2021.723874). As an explanation, it was proposed (Diaz-Rullo and Gonzalez-Pastor, NAR 2023 doi: 10.1093/nar/gkad667) that the observed increase in the capacity to form biofilms in strains that overexpress genes related to Q modification of tRNA would be related to this greater resistance to UV radiation.

We now mention the previous observations suggesting a link between tgt and UV. We thank the referee for the reference which we had overlooked. Note that in the case of our experiments, all cultures are in planktonic form and are not allowed to form biofilms. We thus prefer not to biofilmlinked processes in this study.

**Reviewer #3 (Public Review):**
Summary:In this manuscript the authors begin with the interesting phenotype of sub-inhibitory concentrations of the aminoglycoside tobramycin proving toxic to a knockout of the tRNA-guanine transglycosylase (Tgt) of the important human pathogen, *Vibrio cholerae*. Tgt is important for incorporating queuosine (Q) in place of guanosine at the wobble position of GUN codons. The authors go on to define a mechanism of action where environmental stressors control expression of tgt to control translational decoding of particularly tyrosine codons, skewing the balance from TAC towards TAT decoding in the absence of the enzyme. The authors use advanced proteomics and ribosome profiling to reveal that the loss of tgt results in increased translation of proteins like RsxA and a cohort of DNA repair factors, whose genes harbor an excess of TAT codons in many cases. These findings are bolstered by a series of molecular reporters, mass spectrometry, and tRNA overexpression strains to provide support for a model where Tgt serves as a molecular pivot point to reprogram translational output in response to stress.Strengths:The manuscript has many strengths. The authors use a variety of strains, assays, and advanced techniques to discover a mechanism of action for Tgt in mediating tolerance to sub-inhibitory concentrations of tobramycin. They observe a clear phenotype for a tRNA modification in facilitating reprogramming of the translational response, and the manuscript certainly has value in defining how microbes tolerate antibiotics.

We thank the referee for their time and comments.

Weaknesses:The conclusions of the manuscript are mostly very well-supported by the data, but in some places control experiments or peripheral findings cloud precise conclusions. Some additional clarification, discussion, or even experimental extension could be useful in strengthening these areas.(1) The authors have created and used a variety of relevant molecular tools. In some cases, using these tools in additional assays as controls would be helpful. For example, testing for compensation of the observed phenotypes by overexpression of the Tyrosine tRNA(GUA) in Figure 2A with the 6xTAT strain, Figure 5C with the rxsA-GFP fusion, and/or Figure 7B with UV stress would provide additional information of the ability of tRNA overexpression to compensate for the defect in these situations.

Thank you for the suggestions. Since overexpression of tRNA tyr is not expected to decrease decoding of TAT, we do not necessarily expect any effect for UV and rsxA expression. Overexpression of tRNA_GUA restores fitness of delta-tgt in TOB, but this is probably independent of RsxA. As ref2 also suggested above, we included in the discussion that the effect seen in delta-tgt with TOB is not only due to RsxA expression but also additional processes. However, these suggestions are interesting and we performed the following experiments in order to have an answer for these questions:

- “testing for compensation of the observed phenotypes by overexpression of the Tyrosine tRNA(GUA) in Figure 2A with the 6xTAT strain”:

This is now included in figure S2C and results as follows:

“We also tested decoding reporters for TAT/TAC in WT and ∆tgt overexpressing tRNA-Tyr in trans (Fig. S1C). The presence of the plasmid amplified differences between the two strains with decreased decoding of TAC (and increased TAT, as expected) in ∆tgt with empty plasmid compared to WT. Overexpression of tRNA_TyrGUA did not significantly impact decoding of TAT and increased decoding of TAC as expected. Since overexpression of tRNA_TyrGUA rescues ∆tgt in tobramycin (Fig. 1I) and facilitates TAC decoding, this suggests that issues with TAC codon decoding contribute to the fitness defect observed in ∆tgt upon growth with tobramycin. Overexpression of tRNA_TyrAUA increased decoding of TAT in WT but did not change it in ∆tgt where it is already high. Interestingly, overexpression of TyrAUA also increased decoding of TAC in WT. Thus, overexpression of tRNA_TyrAUA possibly changes the equilibrium between the decoding of TAC vs TAT and may restore translation of TAC enriched transcripts. “

- Figure 5C with the rxsA-GFP fusion:

When we overexpress tRNA_GUA, rsxA fluorescence is 2-fold higher in delta-tgt compared to wt. However, the fluorescence is highly decreased compared to the condition with no tRNA overexpression. While we are not sure whether this apparent decrease is a technical issue or not (e.g. due to the presence of additional plasmid), we prefer not to further explore this in this manuscript. Note that we could not obtain delta-tgt strain carrying both plasmids expressing tRNA_GUA and rsxA, suggesting toxic overproduction of rsxA in this context.

**Author response image 2. sa4fig2:** 

- Figure 7B with UV stress:

Here again, delta-tgt overexpressing tRNA_GUA is still more UV resistant than WT overexpressing tRNA_GUA.

**Author response image 3. sa4fig3:** 

(2) The authors present a clear story with a reprogramming towards TAT codons in the knockout strain, particularly regarding tobramycin treatment. The control experiments often hint at other codons also contributing to the observed phenotypes (e.g., His or Asp), yet these effects are mostly ignored in the discussion. It would be helpful to discuss these findings at a minimum in the discussion section, or possibly experimentally address the role of His or Asp by overexpression of these tRNAs together with Tyrosine tRNA(GUA) in an experiment like that of Figure 1I to see if a more "wild type" phenotype would present. In fact, the synergy of Tyr, His, and/or Asp codons likely helps to explain the effects observed with the DNA repair genes in later experiments.

We thank the referee for the suggestion. We agree that there could be synergies between these codons, and that’s probably why proteomics data does not clearly reflect tyrosine codons usage bias. This is now further discussed in the ideas and speculation section.

Moreover, we have added Figure S3G and the following result:

“Since not all TAT biased proteins are found to be enriched in ∆tgt proteomics data, the sequence context surrounding TAT codons could affect their decoding. To illustrate this, we inserted after the gfp start codon, various tyrosine containing sequences displayed by rsxA (Fig. S3G). The native tyrosines were all TAT codons, our synthetic constructs were either TAT or TAC, while keeping the remaining sequence unchanged. We observe that the production of GFP carrying the TEYTATLLL sequence from RsxA is increased in Δtgt compared to WT, while it is unchanged with TEYTACLLL. However, production of the GFP with the sequences LYTATRLL/LYTACRLL and EYTATLR/ EYTACLR was not unaffected (or even decreased for the latter) by the absence of tgt. Overall, our results demonstrate that RsxA is upregulated in the ∆tgt strain at the translational level, and that proteins with a codon usage bias towards tyrosine TAT are prone to be more efficiently translated in the absence of Q modification, but this is also dependent on the sequence context. “

(3) Regarding Figure 6D, the APB northern blot feels like an afterthought. It was loaded with different amounts of RNA as input and some samples are repeated three times, but Δcrp only once. Collectively, it makes this experiment very difficult to assess.

A different amount of RNA was used only for ∆tgt in which we have only one band because of the absence of modification. For all the other conditions, the same amount of RNA was used (0.9 µg). Additional replicates of crp were in an additional gel but only a representative gel was shown in the manuscript. This is now specified in the legend.

We also attach below the picture of the gel with total RNA (syber Gold labelling of total RNA), where it can be seen that the lanes contain an equivalent quantity of RNA, except for ∆tgt.

**Author response image 4. sa4fig4:** 

Minor Points:(3) Fig S2B, do the authors have a hypothesis why the Asp and Phe tRNAs lead to a growth decrease in the untreated samples? It appears like Phe(GAA) partially compensates for the defect.

Yes we agree, at this stage we do not have any satisfactory answer for this unfortunately. This would be interesting to study further but this is beyond the scope of the present study.

(5) Lines 655 to 660 seem more appropriate as speculation in the discussion rather than as a conclusion in the results, where no direct experiments are performed. The authors might take advantage of the "Ideas and Speculation" section that eLife allows.

Thank you very much for this suggestion, we added this section to the manuscript.

**Recommendations for the authors:**

**Reviewer #1 (Recommendations For The Authors):**
Minor.- Figure 6 - Fonts on several mutants is different size/type. fixed- What is the Pm promoter. Please expand and give enough details so reader can follow. Especially as it is less used in V. cholerae (typical being pBAD or pTAC promoters). done- Spacing where references are inserted should be checked. done- Line 860-863 - "*V. cholerae*'s response to sub-MIC antibiotic stress is transposable to other Gramnegative pathogens" . This reads awkard. Consider rephrasing. done- Figure 7 - Text in A and C is very small and is very hard to read. Font for tgt is different.

Fixed. Tgt is in italics.

**Reviewer #2 (Recommendations For The Authors):**
As specified in the public review, more evidence would be necessary to affirm that tRNAs not modified by Q have a greater preference for translating TAT codons, since there are several previous studies in which it is shown that Q-tRNAs have a greater preference for NAT codons (including TAT). For example, it is suggested to explore what happens with other recoded genes (enriched in TAT or TAC) if there is a high level of Q-tRNAs (overexpression of tgt in a wt context). It is also necessary to clarify how to interpret the Ribo-seq results, which apparently is different from how they have been interpreted in other studies.

Please see above our responses and changes made to the manuscript.

Minor correctionsIn Figure 8, replace "Epitranscriptomic adapation to stress" with "Epitranscriptomic adaptation to stress".

Fixed, thank you for noticing!

**Reviewer #3 (Recommendations For The Authors):**
(1) Lines 48-50, and 110 to 112, the authors have a nice mechanism and story, yet the lines mentioned feel very qualified (e.g., "possibly", "plausibly") and lead to the abstract hiding the value and major conclusions of the study. The authors could consider to revise or even remove these lines to focus on the take-home message in the abstract and end of introduction/discussion.

Thank you for this comment, we modified the text.

(2) Additional description for the samples in the results section for Figure 1 would be helpful to the reader.

Done.

(3) Figure S1, the line of experiments with rluF is interesting, but in the end the choice seems a little random. Have the authors assessed knockouts of other modifications on the ASL for effects? Since the modification is not well characterized in *V. cholerae* according to the authors, it might make sense to save this for a future paper.

We removed S1, as we agree that this experiment does not really add something to the paper.

(4) Line 334 and 353 are redundant.

Fixed.

(5) It is likely beyond the scope of the study, but it would strengthen the paper to repeat Figure 3 with His and/or Asp based on the findings of 2C and 4E to better understand the contribution of His and Asp to Q biology.

We repeated figure 3 with Asp. Based on Fig 2C (less efficient decoding of GAC in deta-tgt in TOB) and 4E (positive GAT codon bias in proteins up in riboseq in delta-tgt TOB), we would expect that beta-lactamase with asp GAC would be less efficiently decoded than GAT in delta-tgt.

This was added to the manuscript

“Like Tyr103, Asp129 was shown to be important for resistance to β-lactams (Doucet et al., 2004; Escobar et al., 1994; Jacob et al., 1990). When we replaced the native Asp129 GAT with the synonymous codon Asp129 GAC, the GAC version did not appear to produce functional β-lactamase in ∆tgt (Fig. 3B), suggesting increased mistranslation or inefficient decoding of the GAC codon by tRNAAsp in the absence of Q. Decoding of GAT codon was also affected in ∆tgt in the presence of tobramycin.”

Added figure: Figure 3B

(6) The authors could consider replacing 5D with S4A-D, which is easier to understand in our opinion.

Done.